EMBO
Molecular Medicine

# Macrophages induce malignant traits in mammary epithelium via IKKε/TBK1 kinases and the serine biosynthesis pathway

Ewa Wilcz-Villega[1], Edward Carter[2,†], Alastair Ironside[2,†], Ruoyan Xu[1,†], Isabella Mataloni[1], Julie Holdsworth[2], William Jones[1], Rocío Moreno Béjar[1], Lukas Uhlik[1], Robert B Bentham[3], Susana A Godinho[1], Jesmond Dalli[4], Richard Grose[2], Gyorgy Szabadkai[3,5,6] [ID], Louise Jones[2], Kairbaan Hodivala-Dilke[2] [ID] & Katiuscia Bianchi[1,*] [ID]

## Abstract

During obesity, macrophages infiltrate the breast tissue leading to low-grade chronic inflammation, a factor considered responsible for the higher risk of breast cancer associated with obesity. Here, we formally demonstrate that breast epithelial cells acquire malignant properties when exposed to medium conditioned by macrophages derived from human healthy donors. These effects were mediated by the breast cancer oncogene IKKε and its downstream target—the serine biosynthesis pathway as demonstrated by genetic or pharmacological tools. Furthermore, amlexanox, an FDA-approved drug targeting IKKε and its homologue TBK1, delayed *in vivo* tumour formation in a combined genetic mouse model of breast cancer and high-fat diet-induced obesity/inflammation. Finally, in human breast cancer tissues, we validated the link between inflammation–IKKε and alteration of cellular metabolism. Altogether, we identified a pathway connecting obesity-driven inflammation to breast cancer and a potential therapeutic strategy to reduce the risk of breast cancer associated with obesity.

**Keywords** inflammation; macrophages; malignant transformation; obesity; tumour metabolism

**Subject Categories** Cancer; Immunology; Metabolism

## Introduction

Obesity is a known risk factor for triple-negative breast cancer in pre-menopausal women (Yang *et al*, 2011) and for oestrogen receptor-positive (ER[+]) breast cancer in post-menopausal women (Agnoli *et al*, 2010). The mechanism responsible for obesity-associated cancer risk is not well established. While the systemic effects of obesity, such as insulin resistance and dysregulated steroid hormones, have been studied extensively, less is known about the local consequences of obesity. In obese women, macrophages infiltrate the breast tissue in the absence of tumours (Sun *et al*, 2012), and both in humans and in mouse models of obesity, the presence of crown structures, consisting of macrophages and necrotic adipocytes, has been reported (Morris *et al*, 2011; Subbaramaiah *et al*, 2011). Macrophages are recruited and activated by monocyte chemoattractant protein 1 (MCP1, also known as CCL2), secreted by adipocytes during obesity, and are necessary for tumour progression (Arendt *et al*, 2013); accordingly, loss of MCP1 delays mammary tumourigenesis in a triple-negative breast cancer model (Cranford *et al*, 2017). Macrophages are thus crucial players in inducing the low level of chronic inflammation associated with obesity that has been proposed to drive malignant transformation at the tissue level (Olson *et al*, 2017), raising the possibility that anti-inflammatory drugs could be used to reduce breast cancer recurrence (Bowers *et al*, 2014). However, obesity-associated inflammation also leads to increased angiogenesis (Arendt *et al*, 2013) and alters extracellular matrix stiffness (Seo *et al*, 2015). In conclusion, the cellular mechanisms by which macrophage-mediated inflammation promotes malignant transformation are poorly characterized.

1   Centre for Molecular Oncology, Barts Cancer Institute, John Vane Science Centre, Queen Mary University of London, London, UK
2   Centre for Tumour Biology, Barts Cancer Institute, John Vane Science Centre, Queen Mary University of London, London, UK
3   Department of Cell and Developmental Biology, Consortium for Mitochondrial Research, University College London, London, UK
4   Lipid Mediator Unit, Biochemical Pharmacology, William Harvey Research Institute, Barts and the London School of Medicine, Queen Mary University of London, London, UK
5   Francis Crick Institute, London, UK
6   Department of Biomedical Sciences, University of Padua, Padua, Italy
    *Corresponding author. Tel: +44 (0)20 7882 2049; E-mail: k.bianchi@qmul.ac.uk
    †These authors contributed equally to this work

Interestingly, the non-canonical members of the IKK family, IκB kinase ε (IKKε) and TANK binding kinase 1 (TBK1) are overexpressed in the white adipose tissue of mice on high-fat diet (HFD) (Chiang et al, 2009; Zhao et al, 2018) and IKKε is induced as a consequence of macrophage infiltration (Sanada et al, 2014). While the role of TBK1 in breast cancer is controversial (Yang et al, 2013; Deng et al, 2014), IKKε is an established breast cancer oncogene, overexpressed in 30% of breast cancer cases (Boehm et al, 2007). IKKε was originally identified as key mediator of the innate immune response (tenOever et al, 2007) and is also known as inducible IKK (IKK-i), being upregulated by several cytokines, e.g. TNF-α, IL6 and IFNγ (Shimada et al, 1999). The IKBKE gene is located on chromosome 1q, which is frequently amplified in breast cancer, partly explaining overexpression of the kinase. However, in around 50% of the cases, the IKBKE transcript is increased (> 2-fold) even in the absence of copy-number changes in its chromosomal region 1q32 (Boehm et al, 2007). Interestingly, in triple-negative breast cancer cell lines without amplification of the IKBKE gene locus, IKKε expression is induced by cytokines, indicating that inflammation could be responsible for IKKε overexpression in the absence of genetic alterations (Barbie et al, 2014).

We thus postulated that during obesity, macrophage infiltration in the breast induces IKKε expression, ultimately contributing to malignant transformation. We show that medium conditioned by macrophages derived from 25 human healthy donors induces acquisition of a malignant phenotype in different 3D systems: (i) the non-transformed MCF10A cells, (ii) mouse primary organoids and (iii) human primary breast epithelial cells derived from patients. Moreover, using pharmacological tools and CRISPR-Cas9 technology, we demonstrate that inhibition of IKKε and its downstream signalling prevents this phenotype. Finally, we show that amlexanox, an FDA-approved drug targeting IKKε and its homologue TBK1, delays tumour appearance in vivo in a combined genetic mouse model of breast cancer and diet-induced obesity. Thus, we have described a signalling pathway linking inflammation and cancer initiation and have identified inhibitors with the potential to reduce the risk of breast cancer in obese patients.

# Results

## Macrophage-conditioned medium induces acquisition of malignant properties

To investigate the consequences of macrophage infiltration in the breast tissue, we used medium conditioned by human peripheral blood mononuclear cells (PBMCs) differentiated and activated as described below. Macrophages show a wide range of phenotypes, influenced by the surrounding microenvironment, but the spectrum of different phenotypes can be characterized into two major groups, such as the classically activated M1 (considered as pro-inflammatory) and alternatively activated M2 macrophages (considered as anti-inflammatory; Murray & Wynn, 2011). We used (i) GM-CSF to induce the differentiation of monocytes to M1-like macrophages (M1D) that were then activated with LPS/IFNγ (M1A) and (ii) M-CSF to induce the differentiation to M2-like macrophages (M2D) that were then activated with IL-4 (M2A) (Fig EV1A). PBMCs were isolated from 25 healthy donors (Fig EV1A–D), and each donor was

labelled with a corresponding letter D (D1–D25), to follow the correlation between each donor and the induced phenotypes. Characterization of the four cell populations via ELISA and cytokine array showed that some markers were shared, such as secretion of MCP1 (Fig EV1E and F), while others were more specific for M1A such as secretion of TNF-α (Fig EV1C), MIG and RANTES (Fig EV1E, G, H) or M2, such as secretion of CCL22 (M2A) (Fig EV1D), IL-10 and TGF-β1 (M2D/A) (Fig EV1E, I, J) (Table EV1). With regard to expression markers known to be induced by certain stimuli (Georgouli et al, 2019) (and associated with certain macrophage subgroups), we observed by FACS analysis that the majority of the cells in the four populations were double-positive for HLA-DR/CD86 (Fig EV2A and B), with M1D/A expressing HLA-DR at higher level than M2A/D (Fig EV2C), while CD86 was higher in M1A and M2A than in M1D and M2D (Fig EV2D). The percentage of cells double positive for CD206/CD163 was higher in the M2D/A populations (Fig EV2E and F), and CD206 was expressed at higher level by M1D and M2A, while CD163 by M2D/A (Fig EV2G and H).

To evaluate the effect of inflammation on epithelial cells, we decided to use the non-transformed breast epithelial cell line MCF10A, as first model system of receiving cells. The ability of cells to grow in an anchorage-independent manner is considered a sign of malignant transformation (Borowicz et al, 2014); thus, we seeded MCF10A cells in soft agar and monitored colony formation over a period of 5 weeks. As a control, we used medium recycled from MCF10A cells themselves, which was compared to media conditioned by M1-differentiated (M1D), M1-activated (M1A), M2-differentiated (M2D) and M2-activated (M2A) macrophages (Fig EV1A and Table EV1). As expected, in control conditions we did not observe any colony formation; however, the four macrophage-conditioned media-induced transformation of MCF10A cells as demonstrated by the presence of colonies with diameters exceeding 50 μm (Fig 1A and EV3A).

We then tested the effect of the macrophage-conditioned media using a standard proliferation assay and observed that M2A, together with M2D and M1D macrophage-conditioned media, also enhanced MCF10A proliferation rate in 2D, but the medium conditioned by M1A macrophages had the opposite effect (Fig EV3B and C), indicating separate mechanisms by which the macrophage-conditioned media affect transformation and proliferation. These discrepancies could be due to the differences between 2D and 3D systems, as previously reported for Ha-Ras and Her2 (Janda et al, 2002), or to different timeframes used in the experiments.

When cultured in 3D conditions, MCF10A cells form acini recapitulating numerous features of the glandular epithelium in vivo (Debnath & Brugge, 2005) and therefore are considered as a physiologically more appropriate model to monitor alterations associated with different stages of tumourigenesis. Thus, in the following experiments, we used this model system to understand the effect of macrophages on epithelial cells.

A hallmark of early tumourigenesis in breast cancer is the displacement of cancer cells from their normal matrix niche and subsequently filling the luminal space of the normally hollow glandular structures (Schafer et al, 2009). To confirm the transforming properties of the different media conditioned by macrophages, we applied them on 16-day-old acini for 24 h. As expected, acini treated with control medium maintained acini-like structures and a clear lumen. On the contrary, all four macrophage-conditioned media (M1D, M1A, M2D and M2A) induced a dramatic increase in the

percentage of acini with a partially or completely filled lumen (Figs 1B and C, and EV3D). Clearance of the lumen during the development of normal spheroids is the result of apoptosis following the detachment of cells from the extracellular matrix; thus, filling of the lumen can be the result of inhibition of cell death (Debnath & Brugge, 2005). However, we applied conditioned media at day 16, when the lumen had already been cleared and the cells were no longer proliferating; thus, we considered two other mechanisms to explain filling of the lumen, such as escape from proliferative arrest (Muthuswamy et al, 2001) and/or cell translocation in combination with acquisition of anchorage-independent survival properties

(Leung & Brugge, 2012). To assess proliferation, we determined the expression of Ki67 (Scholzen & Gerdes, 2000) by immunofluorescence. The percentage of Ki67-positive cells was significantly increased in spheroids treated with M1D- and M1A-conditioned media, but not in M2D or M2A, indicating that M1 and M2 macrophages induce filling of the lumen via different mechanisms (Fig 1D and E). Since Rac activation has been implicated in anchorage-independent survival (Zahir et al, 2003), we took advantage of the Rac1 inhibitor NSC23766 (Gao et al, 2004) and tested its involvement in the macrophage-conditioned medium-induced filling of the lumen. Interestingly, Rac1 inhibition partially prevented this phenotype in

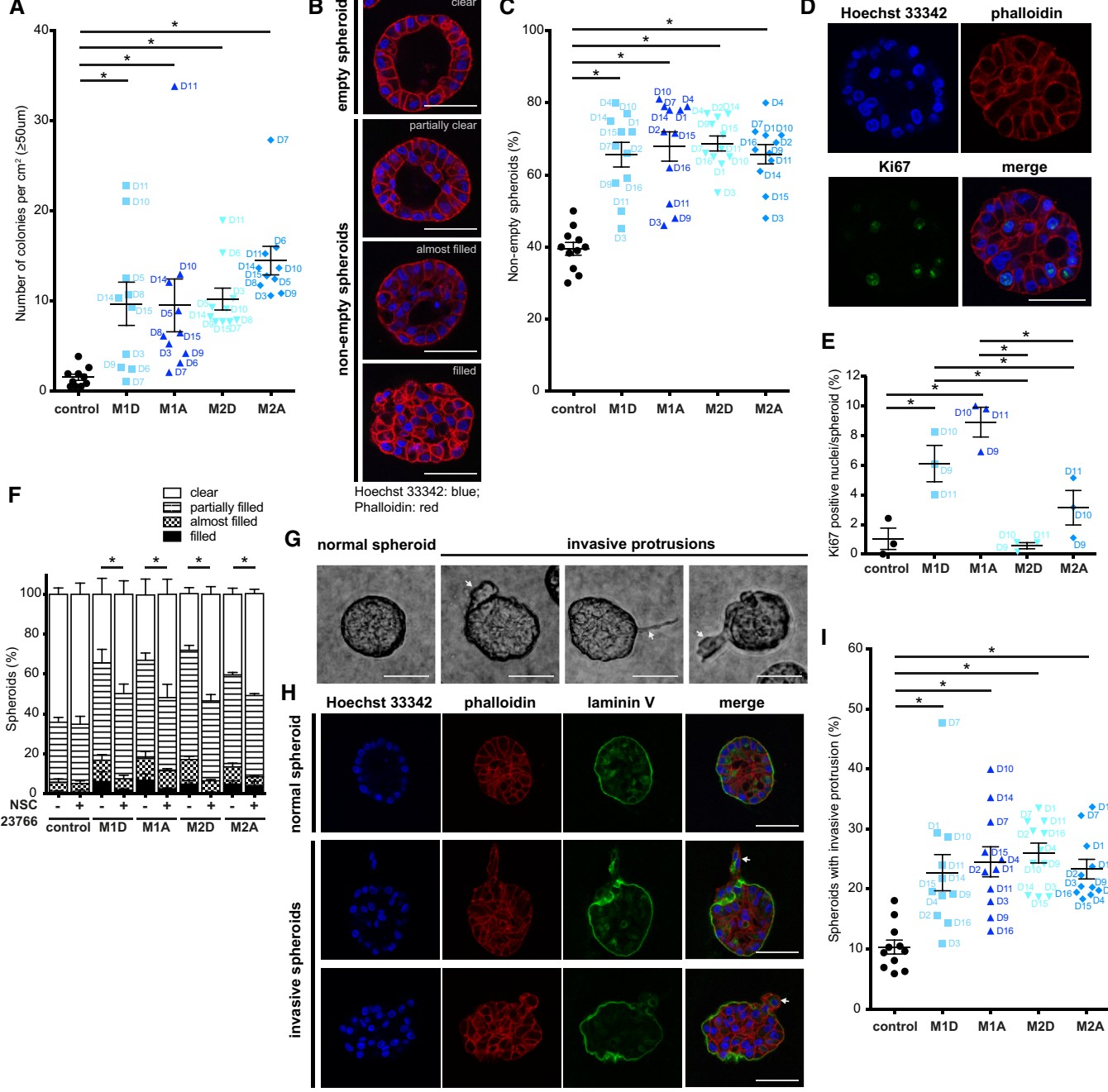

**Figure 1.**

◀

**Figure 1.  Macrophage-conditioned medium promotes anchorage-independent growth and invasiveness of MCF10A cells.**

Sixteen-day-old MCF10A spheroids grown in Matrigel/collagen mix were stimulated for 24 h with either macrophage-conditioned or control medium (B–I).

A   Macrophage-conditioned medium induces MCF10A colony formation in soft agar as compared to control medium. Colonies ≥ 50 μm were counted at 5 weeks. Lines and error bars represent mean ± SEM from 10 independent experiments ($n$ = 3 per condition). Data are shown also in Figs 4A and 6A.

B   Representative images of MCF10A spheroids categorized into four groups according to the filling of the lumen with cell nuclei (clear, partially filled, almost filled and filled). Spheroids stained for DNA (Hoechst 33342 in blue) and F-actin (phalloidin in red).

C   Macrophage-conditioned medium induces filling of the spheroid lumen with cell nuclei compared to control. Lines and error bars represent mean ± SEM from 11 independent experiments ($n$ = 2 per condition; 50 spheroids each). Data are shown also in Figs 4D and 6D.

D   Representative images of Ki67-positive nuclei of MCF10A cells. Spheroids stained for DNA (Hoechst 33342 in blue), F-actin (phalloidin in red) and Ki67 (green).

E   Quantitative analysis of Ki67-positive nuclei per spheroid indicates enhanced proliferation of MCF10A cells upon stimulation with M1 macrophage-conditioned media. Lines and error bars represent mean ± SEM from three independent experiments (20–30 non-empty spheroids per condition).

F   NSC23766 (Rac1 inhibitor; 50 μM) reduces the filling of the spheroid lumen with cell nuclei upon macrophage-conditioned medium stimulation compared to control. Filling of the spheroid lumen with cell nuclei categorized into four groups (clear, partially filled, almost filled and filled). Lines and error bars represent mean ± SEM from three independent experiments ($n$ = 2 per condition; 50 spheroids each). Partially filled, almost filled and filled MCF10A spheroids were combined together (non-empty spheroids) for statistical analysis. All the data shown without the use of NSC23766 are also included in Fig EV3D.

G   Invasive protrusions of spheroids into Matrigel/collagen mix marked with a white arrow.

H   3D structures of spheroids stained for DNA (Hoechst 33342 in blue), F-actin (phalloidin in red) and laminin V (green). Laminin V staining indicates the loss of basal membrane continuity at invasive protrusions. Invasive protrusions are marked with a white arrow.

I   Macrophage-conditioned medium induces invasive protrusions in MCF10A spheroids compared to control. Lines and error bars represent mean ± SEM from 11 independent experiments ($n$ = 2 per condition; at least 15 spheroids each from at least 2 fields of view). Data are shown also in Figs 4E and 6E.

Data information: Macrophage donors are indicated as D1–D16. M1D—M1-differentiated, M1A—M1-activated, M2D—M2-differentiated, M2A—M2-activated macrophages. *$P$ < 0.05 as measured by one-way ANOVA with uncorrected Fisher's LSD *post hoc* test (exact $P$ values are shown in Table EV3). Scale bar: 50 μm. Source data are available online for this figure.

all conditions, indicating that this step is common in the signalling induced by both M1 and M2 macrophages (Figs 1F and EV3E).

Finally, we tested another Rac1-dependent process underlying malignant transformation, i.e. the formation of invasive protrusions accompanied by degradation of laminin V, previously reported in spheroids overexpressing HER2 and stimulated with tumour growth factor β (TGFβ; Wang *et al*, 2006). Culturing 16-day-old MCF10A spheroids for 24 h in all four macrophage-conditioned media induced this invasive phenotype in agreement with Wolford *et al* (2013), typically resulting in one invasive protrusion per spheroid (Fig 1G–I). The effect was blocked by the Rac1 inhibitor NSC23766, as previously reported (Godinho *et al*, 2014) (Fig EV3F), with exception of M1A-conditioned medium. This phenotype was partially reproduced when testing MCF10A cell migration in 2D cultures, using a wound-healing assay. To exclude differential effects on cell proliferation, we performed this assay in the presence of mitomycin C. We showed that M1D-, M2D- and M2A-conditioned media induced a higher migration rate (wound confluency density ~80% versus ~60% of control medium at 48 h), while M1A-conditioned medium reduced the migration rate of MCF10A cells (wound confluency density around ~40% at 48 h; Fig EV3G and H).

Altogether, these data demonstrated that media conditioned by macrophages derived from healthy donors promote acquisition of fundamental malignant properties, such as anchorage-independent growth and invasiveness, in non-tumourigenic breast epithelial MCF10A cells grown in 3D. Of note, the strength of inducing different phenotypes varied between individual donors. For example, medium conditioned by D11-derived M1A induced the highest number of colonies in soft agar (Fig 1A), but had a weak effect in inducing invasive phenotype of the spheroids (Fig 1I).

**Macrophage-conditioned medium induces malignant properties in primary mouse and human model systems**

We then tested the effect of macrophage-conditioned media on mouse primary mammary organoids, a system that recapitulates main traits of the breast acini, where cells are organized in a bilayered structure of basal myoepithelial cells and internal luminal epithelial cells (Nguyen-Ngoc *et al*, 2015). Importantly, we confirmed formation of invasive protrusions in mouse primary organoids cultured in M1A- or M2A-conditioned medium (M1D and M2D were not tested on organoids) (Fig 2A and B). The staining for α-SMA revealed the loss of bilayered structure, with luminal cells found commonly at the front of invasive protrusions (Fig 2C). Macrophages infiltrate breast tissue early during obesity and are considered as inducing low level of chronic inflammation locally (Sun *et al*, 2012). Because this observation is recapitulated in a diet-induced model of obesity in mice (Arendt *et al*, 2013), we tested the invasive behaviour of mouse primary mammary organoids derived from animals on normal diet (ND) vs. high-fat diet (HFD) (Fig EV4A). As expected, animals on HFD were heavier than the ones on ND (average body weight: 27.2 ± 1.48 g vs. 21.3 ± 1.27 g, respectively), adipocytes in the mammary fat pad were on average larger (major adipocyte diameter: 74.4 ± 17.1 μm vs. 42.5 ± 7.7 μm), and we could observe a higher number of infiltrating macrophages (Fig EV4B–F), as previously reported (Subbaramaiah *et al*, 2011; Incio *et al*, 2018). When organoids were seeded in collagen, a substrate which promotes collective invasion (Nguyen-Ngoc *et al*, 2015), organoids derived from mice on HFD were more invasive than the ones derived from mice of ND (Fig 2D–F), suggesting that the environment created by HFD affects the invasive properties of the cells, via a mechanism that is conserved when the cells are cultured *in vitro*.

Altogether, these data demonstrated that media conditioned by macrophages derived from healthy donors promote acquisition of malignant properties in breast primary organoids. Moreover, HFD affects the invasive properties of these cells. Importantly, M1A- or M2A-conditioned medium also induced filling of the lumen—recapitulating ductal carcinoma *in situ*—in a recently established model where myoepithelial and luminal cells, isolated from reduction mammoplasty, are grown in collagen and reform into bilayer structures (Fig 2G and H), a phenotype induced also by overexpression of HER2 (Carter *et al*, 2017).

## IKKε expression is associated with tumour inflammation and mediates the macrophage-conditioned medium-induced malignant phenotype

The breast cancer oncogene IKKε is a kinase linking obesity and inflammation, and inflammation and cancer (Olefsky, 2009); thus, we tested its role in inflammation-mediated transformation of breast epithelial cells. First, to test the association between IKKε expression and tumour inflammation, we performed semi-quantitative analysis of IKKε expression and immune cell infiltration by immunohistochemistry in a cohort of 66 human breast cancers (Table EV2). We observed that in 83% of cases, the kinase was expressed at medium or high level (Fig 3A and B), which showed strong correlation with

the inflammatory grade (Spearman's correlation coefficient 0.619, $P < 0.0001$) (Figs 3C and EV5A). Interestingly, the level of expression of IKKε was distributed heterogeneously in tumours, with higher expression in small tumour cell patches in the stroma and at the invasive front (Fig EV5B and C). Moreover, transcriptomic analysis of the METABRIC breast cancer dataset (Curtis et al, 2012) confirmed that the transcript levels of IKBKE mRNA and the immune infiltration-associated gene expression pattern (immune score, ESTIMATE; Yoshihara et al, 2013) are correlated (Fig 3D), while the IKBKE gene was amplified only in a few cases (Fig 3E). Importantly, the expression of TBK1 showed no correlation with the immune score (Fig EV5D), indicating the specific involvement of IKKε in the response.

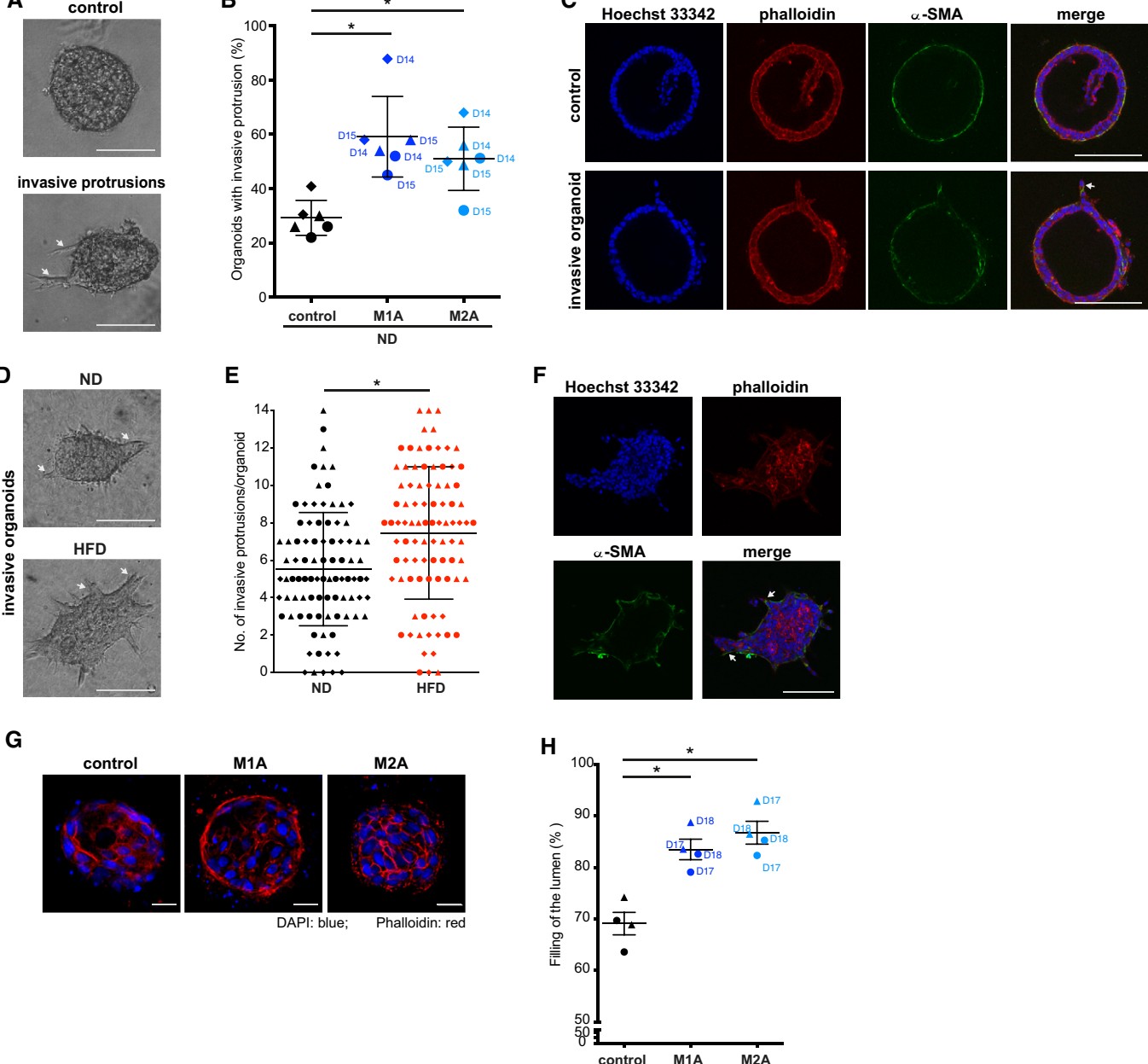

**Figure 2.**

◀ **Figure 2. Macrophage-conditioned medium promotes acquisition of malignant properties in mouse mammary organoids and human ductal structures.**

Mammary organoids were isolated from 19- to 20-week-old C57Bl/6 mice on normal (ND) or high-fat diet (HFD) (n = 3) (A–F).

A–C  Organoids isolated from ND mice were cultured in Matrigel/collagen mix for 2 days followed by macrophage-conditioned or control medium stimulation for 24 h. (A) Representative images of organoids in Matrigel/collagen mix. (B) Macrophage-conditioned medium induces invasive protrusions in organoids as compared to control. Lines and error bars represent mean ± SEM from three independent experiments, with each mouse labelled with a different symbol shape (n = 2 per condition; 15–25 organoids each). *P < 0.05 as measured by one-way ANOVA with uncorrected Fisher's LSD *post hoc* test. Data are shown also in Figs 4F and 6F. (C) 3D structure of organoids stained for DNA (Hoechst 33342 in blue), F-actin (phalloidin in red) and α-SMA (green). Bilayered structure of internal luminal cells and external basal myoepithelial cells is established for non-invasive organoids.

D–F  Organoids isolated from ND or HFD mice were cultured in collagen for 2 days. (D) Representative images of organoids cultured in collagen for 2 days. (E) The number of invasive protrusions per organoid is higher for organoids isolated from mice on HFD compared to mice on ND. Lines and error bars represent mean ± SD from three independent experiments where 30 organoids were counted per each mouse (labelled with a different symbol shape). *P < 0.05 as measured by two-tailed Student's *t*-test. Data are shown also in Figs 4G/H and 6G/H. (F) 3D structure of organoid stained for DNA (Hoechst 33342 in blue), F-actin (phalloidin in red) and α-SMA (green). α-SMA, myoepithelial cell marker, appeared to be loss at invasive protrusions.

G, H  Human myoepithelial and luminal cells isolated from breast specimens were cultured in collagen gels for 14 days to reform ductal structures with luminal compartment. Reformed ducts were then cultured for 7 days in macrophage-conditioned or control medium. (G) Representative images of ductal structures cultured with conditioned medium. Ducts were stained for DNA (DAPI in blue) and F-actin (phalloidin in red). (H) Macrophage-conditioned medium induces filling of the duct lumen with cell nuclei compared to control. Lines and error bars represent mean ± SEM (a total of 24 structures were counted per each condition: 2 patient ductal structures (n = 2), each labelled with a different symbol shape, were cultured with macrophage-conditioned media from either donor 17 or donor 18, 3 ducts each). Filling of the lumen was determined as % of luminal space filled with cells. *P < 0.05 as measured by one-way ANOVA with uncorrected Fisher's LSD *post hoc* test.

Data information: Macrophage donors are indicated as D14, D15, D17, D18. M1A—M1-activated, M2A—M2-activated macrophages. Data are shown also in Figs 4J and 6J. Exact P values are shown in Table EV3. Invasive protrusions are marked with a white arrow. Scale bar: 100 μm (A, C, D, F) or 20 μm (G).

Source data are available online for this figure.

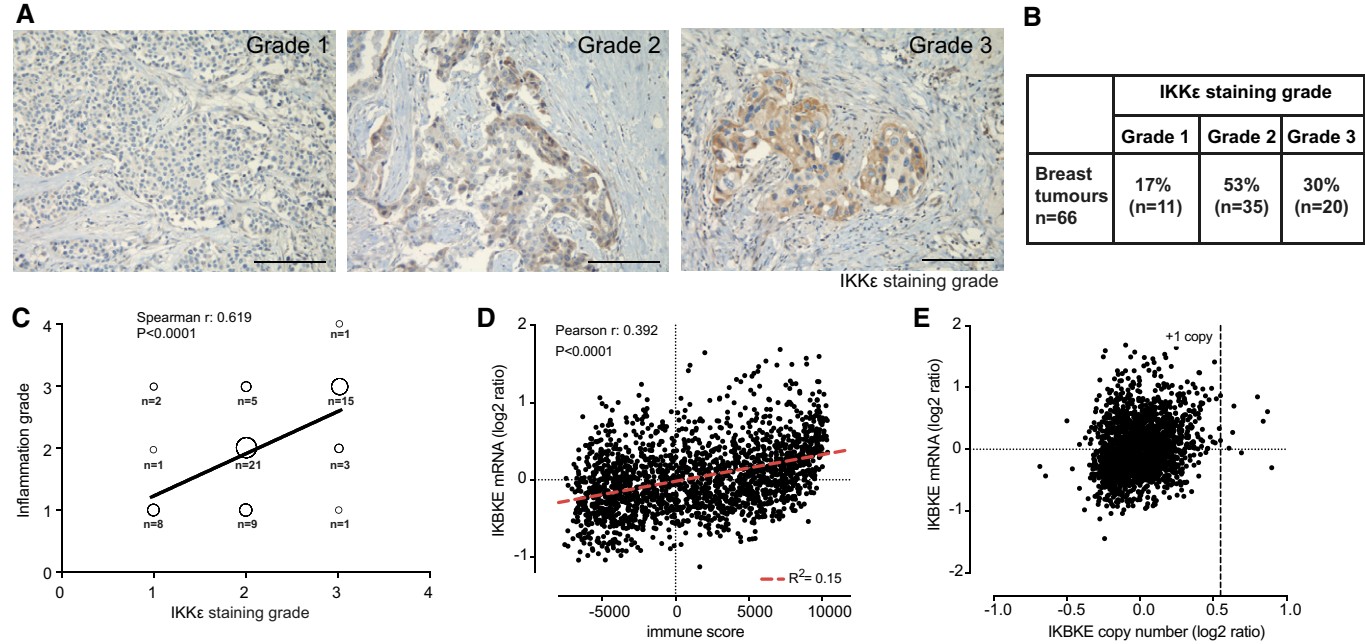

**Figure 3. IKKε expression in breast cancer is associated with immune infiltration.**

A  IKKε expression was assessed semi-quantitatively using a 0–3 grade scale ("0": no staining; "1": weak; "2": moderate; "3": strong) on human breast cancer sections using immunohistochemistry. Representative images according to the IKKε staining grade are shown. Scale bar: 100 μm.

B  Distribution table of IKKε expression in the cohort of 66 human breast carcinomas according to the staining grade.

C  Correlation between IKKε expression and immune cell infiltration ("0": no inflammatory cells; "1": weak; "2": moderate; "3": strong; "4": very strong). Bubble plot of IKKε and immune infiltration grades, size shows the number of tumours falling in the category. Spearman's rho coefficient and significance of difference from slope = 0 (P) are shown.

D  Correlation between IKBKE mRNA levels (log₂ ratio) and immune signature (Yoshihara et al, 2013) in the METABRIC transcriptomic dataset from 1981 breast cancer patients (Curtis et al, 2012). Pearson's correlation rho coefficient and significance of difference from slope = 0 (P) are shown.

E  *IKBKE* gene copy number and mRNA levels (log2 ratio) from the METABRIC transcriptomic dataset (see D) are plotted. Estimated + 1 copy number values are shown by the dotted vertical line. No significant correlation by Pearson's correlation rho coefficient and significance of difference from slope = 0 (P) was detected.

Source data are available online for this figure.

To demonstrate the functional role of IKKε in the macrophage-mediated transformation, we then used the recently identified inhibitor of IKKε/TBK1, amlexanox, an FDA-approved drug previously used for the treatment of mouth ulcers (Reilly *et al*, 2013). Strikingly, when amlexanox 50 μM (as used in ref. Reilly *et al*, 2013) was added to macrophage-conditioned media applied to MCF10A cells, we observed strong inhibition of all cellular transformation-related features, such as anchorage-independent growth (Fig 4A), filling of the spheroid lumen (Fig 4B–D) and formation of invasive protrusions (Fig 4E) induced by M1D-, M1A-, M2A- and M2D-conditioned media. Moreover, amlexanox reduced the percentage of organoids with invasive protrusions induced by M1A- and M2A-conditioned media in mouse primary breast organoids (Fig 4F) as well as the number of invasive protrusions observed in organoids derived from animals on ND and HFD (Fig 4G and H). Filling of the lumen was inhibited by amlexanox also in the human model system where myoepithelial and luminal cells, isolated from reduction mammoplasty, are grown in collagen and reform into bilayer structures (Fig 4I and J). Confirming the crucial role played by IKKε in the phenotypes induced by macrophage-conditioned media, MCF10A cells where the *IKBKE* gene (encoding for IKKε) has been deleted via CRISPR/Cas9 formed dramatically less colonies in soft agar and failed to form invasive protrusions (Figs 4K and L, and EV6A and B).

The effect of amlexanox also in control conditions, both in MCF10A and in organoids, suggests a basal level of IKKε activity.

Altogether, these data indicate that expression of IKKε at the very early stages of transformation is associated with inflammation and amlexanox inhibits acquisition of malignant properties induced by macrophage-conditioned medium.

### NCT502, an inhibitor of PHGDH, prevents macrophage-conditioned medium-induced malignant phenotype

We have recently observed that IKKε regulates the serine biosynthesis pathway (SBP) (https://biorxiv.org/cgi/content/short/855361v1), which previously was reported to support tumour formation and growth via multiple mechanisms (Locasale, 2013). Phosphoglycerate dehydrogenase (PHGDH), the first enzyme of the pathway, is an established oncogene in breast cancer and melanoma (Locasale *et al*, 2011; Possemato *et al*, 2011), and when overexpressed in MCF10A spheroids, it leads to the formation of disorganized spheroids with filled lumen (Locasale *et al*, 2011). To assess whether this pathway is associated with the inflammatory phenotype in breast cancer, we first analysed the level of expression of phosphoserine aminotransferase 1 (PSAT1, second enzyme of the serine biosynthesis pathway) that we recently observed to be regulated by IKKε (https://biorxiv.org/cgi/content/short/855361v1). We

**Figure 4. The macrophage-mediated transforming effect is reduced by amlexanox, an inhibitor of IKKε/TBK1, or upon IKKε deletion.**

A Amlexanox inhibits colony formation of MCF10A cells grown in soft agar with macrophage-conditioned medium for 5 weeks compared to controls. Colonies ≥ 50 μm were counted. Lines and error bars represent mean ± SEM from 10 independent experiments (*n* = 3 per condition). All the data shown without the use of amlexanox are also included in Fig 1A.

B–E Sixteen-day-old MCF10A spheroids grown in Matrigel/collagen mix were stimulated for 24 h with either macrophage-conditioned or control medium in the presence of amlexanox. (B) Representative images showing the filling of the spheroid lumen with cell nuclei. The spheroids were stained for DNA (Hoechst 33342 in blue) and F-actin (phalloidin in red). Scale bar: 50 μm. (C, D) Amlexanox reduces filling of the spheroid lumen with cell nuclei. (C) Filling of the spheroid lumen with cell nuclei categorized into 4 groups (clear, partially filled, almost filled and filled). (D) Filling of the spheroid lumen with cell nuclei. Partially filled, almost filled and filled spheroids were combined together (non-empty spheroids). Lines and error bars represent mean ± SEM from seven independent experiments (*n* = 2 per condition; 50 spheroids each). Partially filled, almost filled and filled spheroids were combined together (non-empty spheroids) for statistical analysis. All the data shown without the use of amlexanox are also included in Figs EV3D and 1C, respectively. (E) Amlexanox reduces the number of spheroids with invasive protrusions. Lines and error bars represent mean ± SEM from eight independent experiments (*n* = 2 per condition, at least 15 spheroids each from at least 2 fields of view). All the data shown without the use of amlexanox are also included in Fig 1I.

F–H Mammary organoids were isolated from 19- to 20-week-old C57Bl/6 mice on normal (ND) or high-fat diet (HFD) (*n* = 3). (F) Organoids isolated from mice on ND were cultured in Matrigel/collagen mix for 2 days and then stimulated with macrophage-conditioned or control medium in the presence of amlexanox. Amlexanox reduces the number of invasive organoids. Lines and error bars represent mean ± SEM from three independent experiments, with each mouse labelled with a different symbol shape (*n* = 2 per condition; 15–25 organoids each). All the data shown without the use of amlexanox are also included in Fig 2B. (G, H) Organoids were cultured in collagen for 2 days after which the number of invasive protrusions per each organoid was determined microscopically. Amlexanox reduces the number of invasive protrusions in organoids isolated from mice on (G) ND or (H) HFD. Lines and error bars represent mean ± SD from 3 independent experiments where 30 organoids were counted per each mouse (labelled with a different symbol shape). All the data shown without the use of amlexanox are also included in Fig 2E.

I, J Human myoepithelial and luminal cells isolated from breast specimens were cultured in collagen gels for 14 days to reform ductal structures with luminal compartment. Reformed ducts were then cultured for 7 days in macrophage-conditioned or control medium in the presence of amlexanox. (I) Representative images showing the filling of the ductal lumen with cell nuclei. The ducts were stained for DNA (DAPI in blue) and F-actin (phalloidin in red). Scale bar: 20 μm. (J) Amlexanox inhibits filling of the duct lumen with cell nuclei. Lines and error bars represent mean ± SEM (a total of 24 structures were counted per each condition: 2 patient ductal structures (*n* = 2), each labelled with a different symbol shape, were cultured with macrophage-conditioned media from either donor 17 or donor 18, 3 ducts each). Filling of the lumen was determined as % of luminal space filled with cells. All the data shown without the use of amlexanox are also included in Fig 2H.

K, L *IKBKE* gene (encoding for IKKε) was deleted in MCF10A cells via CRISPR-Cas9 technology. Three knockout clones were combined for each cell line. (K) IKKε CRISPR-Cas9 knockout MCF10A cells form less colonies when grown in soft agar with macrophage-conditioned medium for 5 weeks compared to CRISPR-Cas9 control (non-targeting crRNA) cells. Colonies ≥ 50 μm were counted. (L) Sixteen-day-old MCF10A spheroids grown in Matrigel/collagen mix were stimulated for 24 h with either macrophage-conditioned or control medium. IKKε CRISPR-Cas9 knockout MCF10A cells are less invasive compared to CRISPR control (non-targeting crRNA) cells. Lines and error bars represent mean ± SEM from 3 independent experiments (*n* = 3 per condition). Media from two macrophage donors were combined in 1:1 ratio in each experiment.

Data information: Macrophage donors are indicated as D1-D25. M1D—M1-differentiated, M1A—M1-activated, M2D—M2-differentiated, M2A—M2-activated macrophages. Aml.—amlexanox (50 μM). *P < 0.05 as measured by two-tailed Student's *t*-test (exact P values are shown in Table EV3).
Source data are available online for this figure.

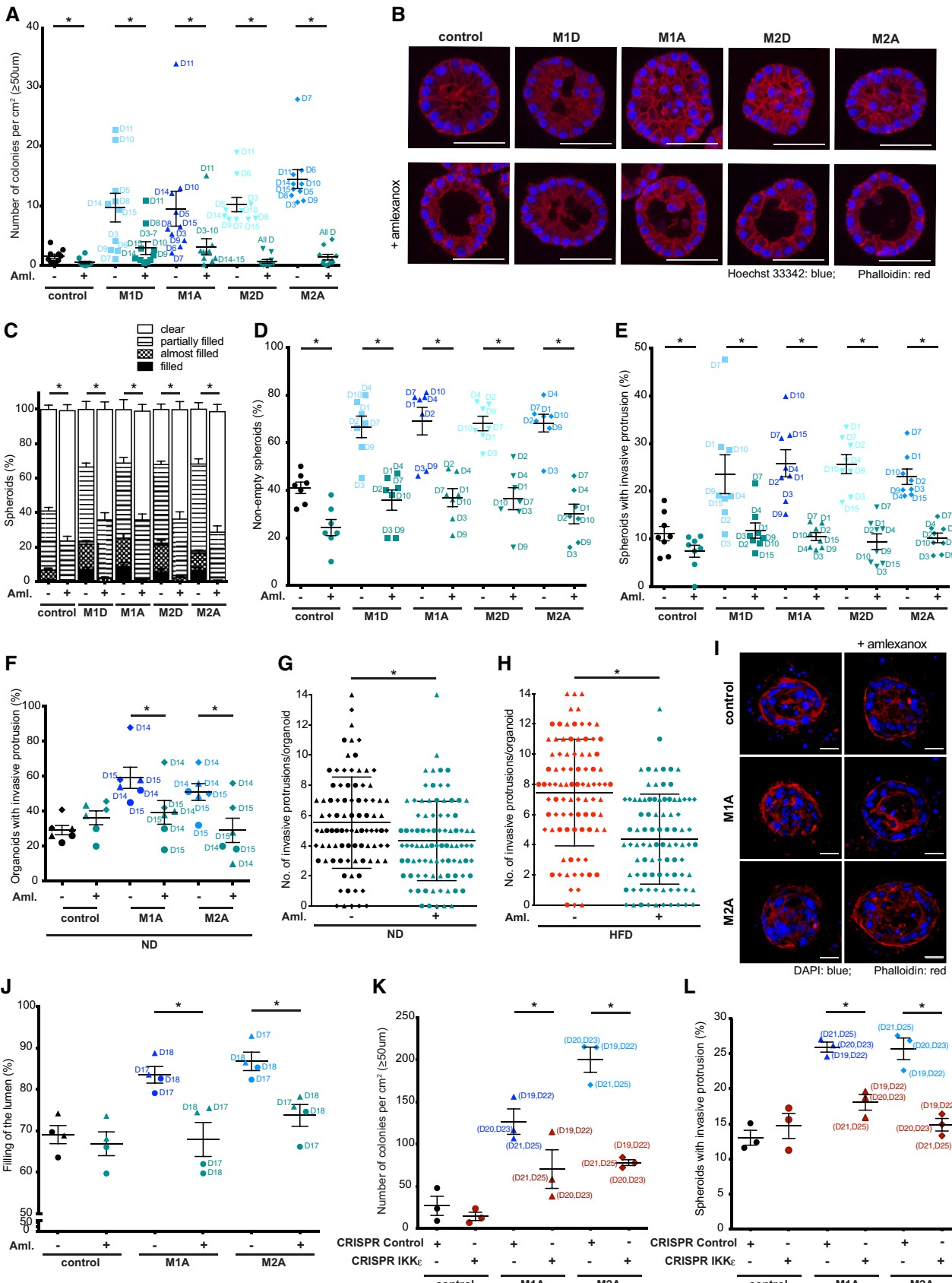

**Figure 4.**

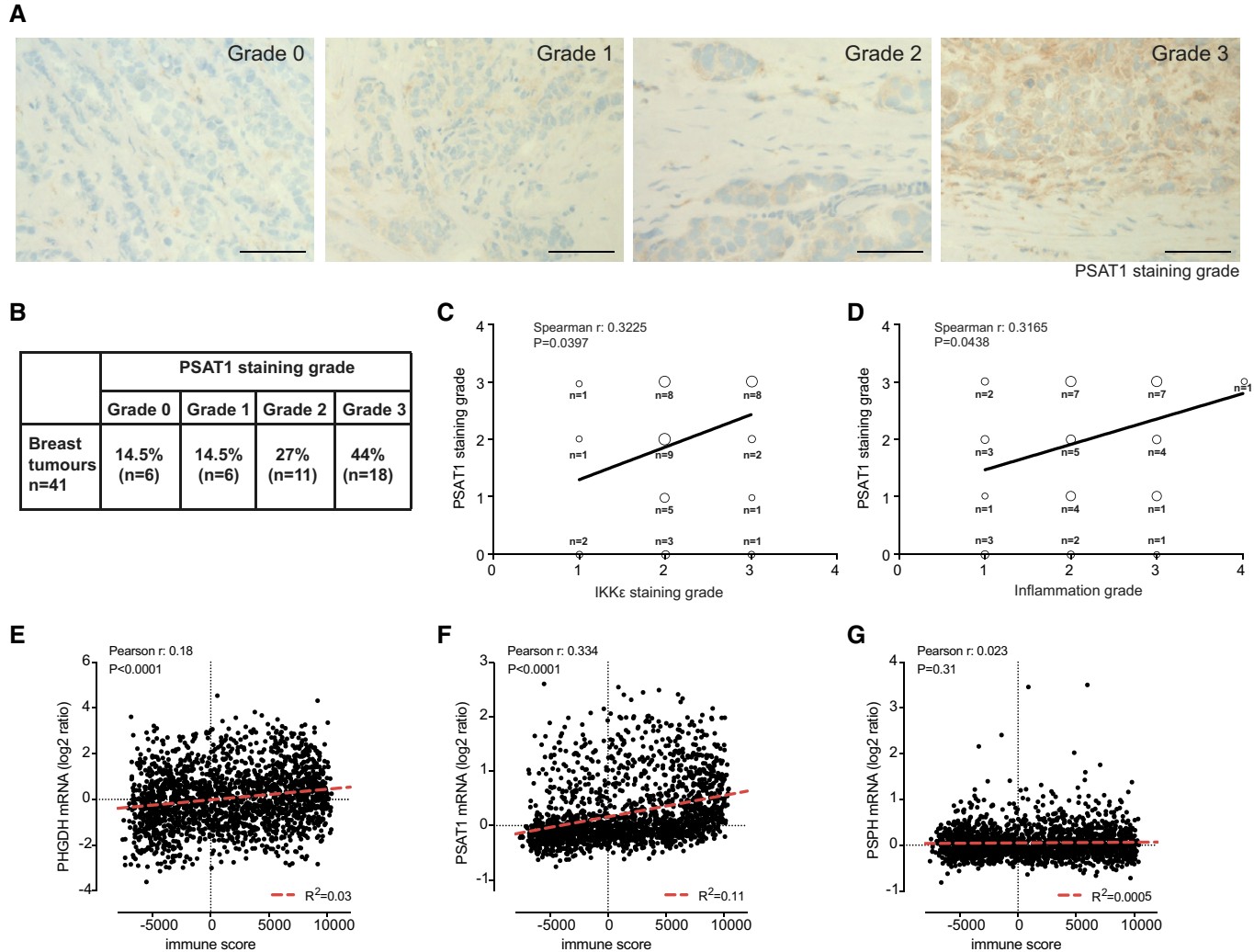

**Figure 5. PSAT1 expression in breast cancer is associated with IKKε expression and immune infiltration.**

A    PSAT1 expression was assessed semi-quantitatively using a 0-3 grade scale ("0": no staining; "1": weak; "2": moderate; "3": strong) on human breast cancer sections using immunohistochemistry. Representative images according to the PSAT1 staining grade are shown. Scale bar: 50 μm.

B    Distribution table of PSAT1 expression in the cohort of 41 human breast carcinomas according to the staining grade.

C    Correlation between PSAT1 and IKKε expression ("0": no staining; "1": weak; "2": moderate; "3": strong). Bubble plot of PSAT1 and IKKε staining grades, size shows the number of tumours falling in the category.

D    Correlation between PSAT1 and immune cell infiltration ("0": no inflammatory cells; "1": weak; "2": moderate; "3": strong; "4": very strong). Bubble plot of PSAT1 and immune infiltration grades, size shows the number of tumours falling in the category.

E–G  Correlation of mRNA levels of the SBP enzyme genes *PSAT1* (E), *PHGDH* (F), *PSPH* (G) with the immune signature (Yoshihara *et al*, 2013) in the METABRIC transcriptomic dataset from 1981 breast cancer patient (Curtis *et al*, 2012).

Data information: Spearman's (C, D) or Pearson's (E–G) correlation rho coefficient and significance of difference from slope = 0 (*P*) are shown.
Source data are available online for this figure.

used frozen sections of 41 patients of the cohort previously used for IKKε staining to semi-quantitatively define PSAT1 expression. We found that in 71% of cases, the protein was expressed at medium or high level (grade 2 or 3, Fig 5A and B). Importantly, we found a significant correlation of PSAT1 expression with IKKε levels and the inflammatory grade of the same samples (Fig 5C and D), indicating association between inflammation and SBP in breast tumours.

We also found significant correlation between the transcript levels of two out of three SBP enzymes—PHGDH and PSAT1—

and the immune infiltration-associated gene expression pattern (immune score, ESTIMATE) (Yoshihara *et al*, 2013) (Fig 5E–G) in the METABRIC breast cancer transcriptome dataset (Curtis *et al*, 2012), indicating that inflammation is also a key regulator of the SBP in addition to known oncogenes (Yang & Vousden, 2016).

These results suggested that activation of the SBP downstream of IKKε could contribute to the acquisition of malignant properties in MCF10A cells induced by macrophage-conditioned medium. To test this hypothesis, we applied NCT502 (10 μM), a recently

developed inhibitor of PHGDH (Pacold *et al*, 2016), to the macro-phage-conditioned media. Similarly to the effect of IKKε inhibition by amlexanox, NCT502 prevented anchorage-independent growth (Fig 6A) and filling of the lumen (Fig 6B–D) induced by M1D-, M1A-, M2A- and M2D-conditioned media. NCT502 also inhibited formation of invasive protrusions induced by M1D, M2A and M2D, but not by M1A (Fig 6E). Furthermore, NCT502 reduced the percentage of organoids with invasive protrusions induced by M1A- and M2A-conditioned media in primary mouse breast orga-noids (Fig 6F) and reduced the number of invasive protrusions in organoids derived from animals on ND and HFD (Fig 6G and H). Importantly, filling of the spheroid lumen was inhibited by NCT502 also in the human model system where bilayer structures are formed in collagen by myoepithelial and luminal cells isolated from reduction mammoplasty (Fig 6I and J). Confirming the crucial role played by the SBP downstream of IKKε, macrophage-conditioned media did not induce any phenotype (i.e. colonies in soft agar and invasive protrusions) in MCF10A cells where the *PHGDH* gene has been deleted via CRISPR/Cas9 (Figs 6K and L, and EV6A and B).

These data indicate that activation of the SBP is essential for the acquisition of the malignant phenotypes induced by macrophage-conditioned media.

## Amlexanox delays tumourigenesis *in vivo* in a genetic breast cancer mouse model combined with diet-induced obesity

It has been previously shown that HFD-induced obesity enhances mammary carcinogenesis in a mouse model of breast cancer carry-ing a heterozygous copy of the mouse mammary tumour virus–poly-omavirus middle T antigen (MMTV-PyMT) (Cowen *et al*, 2015). Thus, we used this model to test whether IKKε plays a role in obesity-induced tumourigenesis.

Six- to seven-week-old female mice (MMTV-PyMT$^{+/-}$, 10 per group) were fed on ND or HFD, and a week after the start of the diet, they received vehicle or amlexanox treatment by daily oral gavage (Fig 7A). A smaller cohort of WT animals were kept on the same diet to control for possible unpredicted effects of the MMTV-PyMT gene. Body weight and tumour appearance were monitored for a total period of 12 weeks (from the first week after beginning of the diet). Wild-type mice on HFD gained significantly more weight (Fig 7B), adipocytes were larger in the mammary fat pad (Fig 7C), and more infiltrating macrophages were observed (Fig 7D). MMTV-PyMT$^{+/-}$ animals also gained more weight on the HFD regime, which was unaffected by amlexanox treatment (Fig 7E–G), differently from what previously reported (Reilly *et al*, 2013). Importantly, however, tumour latency was significantly

---

**Figure 6. The macrophage-mediated transforming effect is reduced by NCT502, PHGDH inhibitor or upon PHGDH deletion.**

A  NCT502 (10 μM) inhibits colony formation of MCF10A cells grown in soft agar with macrophage-conditioned medium for 5 weeks compared to controls. Colonies ≥ 50 μm were counted. Lines and error bars represent mean ± SEM from five independent experiments (*n* = 3 per condition). All the data shown without the use of NCT502 are also included in Fig 1A.

B–E  Sixteen-day-old MCF10A spheroids grown in Matrigel/collagen mix were stimulated for 24 h with either macrophage-conditioned or control medium in the presence of NCT502 (10 μM). (B) Representative images showing the filling of the spheroid lumen with cell nuclei. The spheroids were stained for DNA (Hoechst 33342 in blue) and F-actin (phalloidin in red). Scale bar: 50 μm. (C, D) NCT502 reduces the filling of the spheroid lumen with cell nuclei. (C) Filling of the spheroid lumen with cell nuclei categorized into 4 groups (clear, partially filled, almost filled and filled). (D) Filling of the spheroid lumen with cell nuclei. Partially filled, almost filled and filled spheroids were combined together (non-empty spheroids). Lines and error bars represent mean ± SEM from 3 independent experiments (*n* = 2 per condition; 50 spheroids each). Partially filled, almost filled and filled spheroids were combined together (non-empty spheroids) for statistical analysis. All the data shown without the use of NCT502 are also included in Figs EV3D and 1C, respectively. (E) NCT502 reduces the number of spheroids with invasive protrusions (*n* = 2 per condition). Lines and error bars represent mean ± SEM from three independent experiments (*n* = 2 per condition; at least 15 spheroids each from at least 2 fields of view). All the data shown without the use of NCT502 are also included in Fig 1I.

F–H  Mouse mammary organoids were isolated from 19- to 20-week-old C57Bl/6 mice that were either on normal (ND) or high-fat diet (HFD) (*n* = 3). (F) Organoids were isolated from mice on ND and cultured in Matrigel/collagen mix for 2 days followed by macrophage-conditioned or control medium stimulation for 24 h in the presence of NCT502 (10 μM). NCT502 reduces the number of invasive organoids. Lines and error bars represent mean ± SEM from 3 independent experiments, with each mouse labelled with a different symbol shape (*n* = 2 per condition; 15–25 organoids each). All the data shown without the use of NCT502 are also included in Fig 2B. (G, H) Organoids were cultured in collagen for 2 days after which the number of invasive protrusions per each organoid was determined microscopically. NCT502 (10 μM) reduces the number of invasive protrusions in organoids isolated from mice that were either on (G) ND or (H) HFD. Lines and error bars represent mean ± SD from 3 independent experiments where 30 organoids were counted per each mouse (labelled with a different symbol shape). All the data shown without the use of NCT502 are also included in Fig 2E.

I, J  Human myoepithelial and luminal cells isolated from breast specimens were cultured in collagen gels for 14 days to reform ductal structures with luminal compartment. Reformed ducts were then cultured for 7 days in macrophage-conditioned or control medium in the presence of NCT502 (10 μM). (I) Representative images showing the filling of the ductal lumen with cell nuclei. The ducts were stained for DNA (DAPI in blue) and F-actin (phalloidin in red). Scale bar: 20 μm. (J) NCT502 inhibits filling of the duct lumen with cell nuclei. Lines and error bars represent mean ± SEM (a total of 24 structures were counted per each condition: 2 patient ductal structures (*n* = 2), each labelled with a different symbol shape, were cultured with macrophage-conditioned media from either donor 17 or donor 18, 3 ducts each). Filling of the lumen was determined as % of luminal space filled with cells. All the data shown without the use of NCT502 are also included in Fig 2H.

K, L  *PHGDH* gene was deleted in MCF10A cells via CRISPR-Cas9 technology. Three knockout clones were combined for each cell line. (K) PHGDH CRISPR-Cas9 knockout MCF10A cells form less colonies when grown in soft agar with macrophage-conditioned medium for 5 weeks compared to CRISPR-Cas9 control (non-targeting crRNA) cells. Colonies ≥ 50 μm were counted. (L) Sixteen-day-old MCF10A spheroids grown in Matrigel/collagen mix were stimulated for 24 h with either macrophage-conditioned or control medium. PHGDH CRISPR-Cas9 knockout MCF10A cells are less invasive compared to CRISPR control (non-targeting crRNA) cells. Lines and error bars represent mean ± SEM from 3 independent experiments (*n* = 3 per condition). Media from two macrophage donors were combined in 1:1 ratio in each experiment.

Data information: Macrophage donors are indicated as D1-D25. M1D—M1-differentiated, M1A—M1-activated, M2D—M2-differentiated, M2A—M2-activated macrophages. *$P < 0.05$ as measured by two-tailed Student's *t*-test (exact *P* values are shown in Table EV3).
Source data are available online for this figure.

---

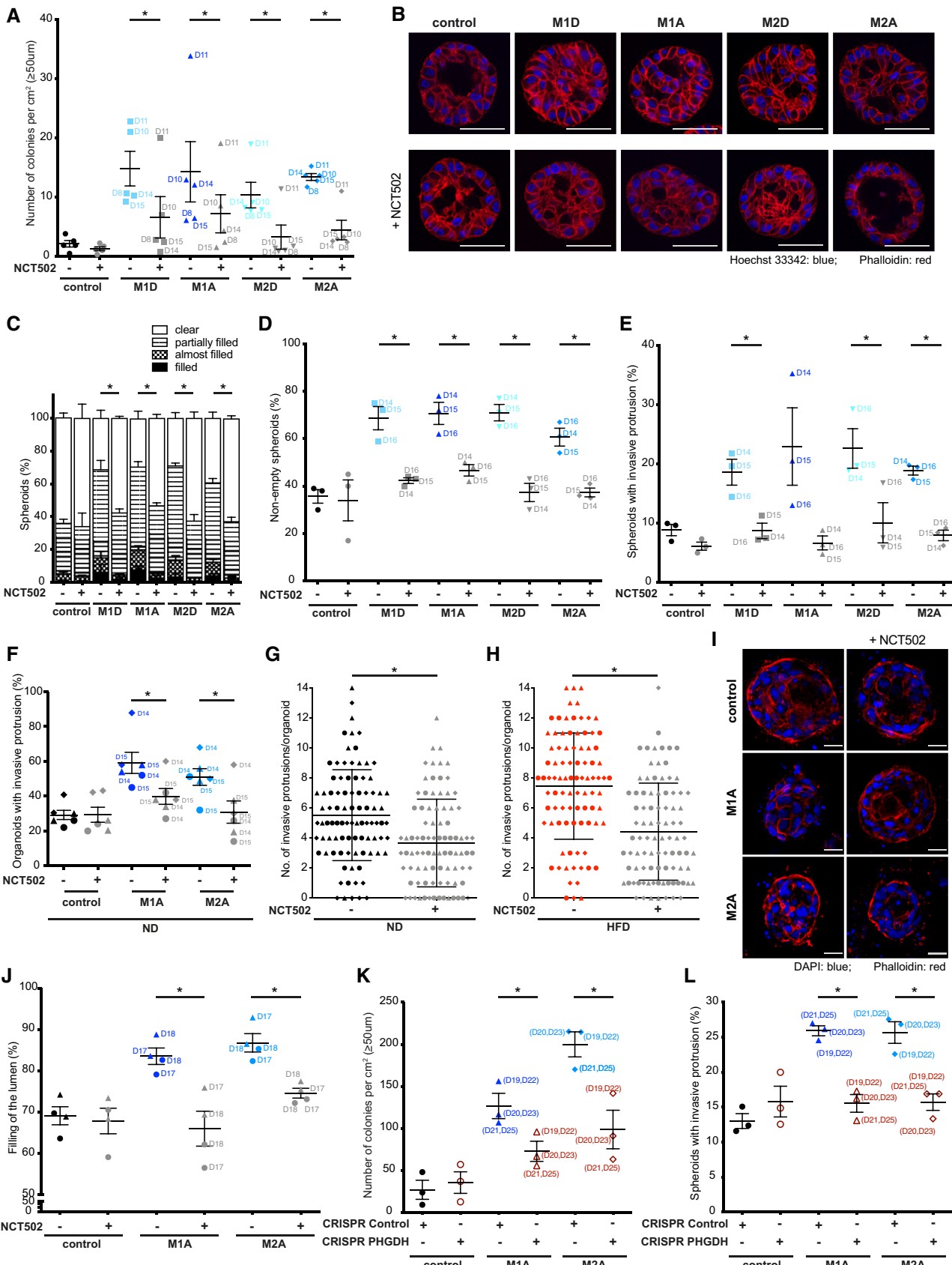

**Figure 6.**

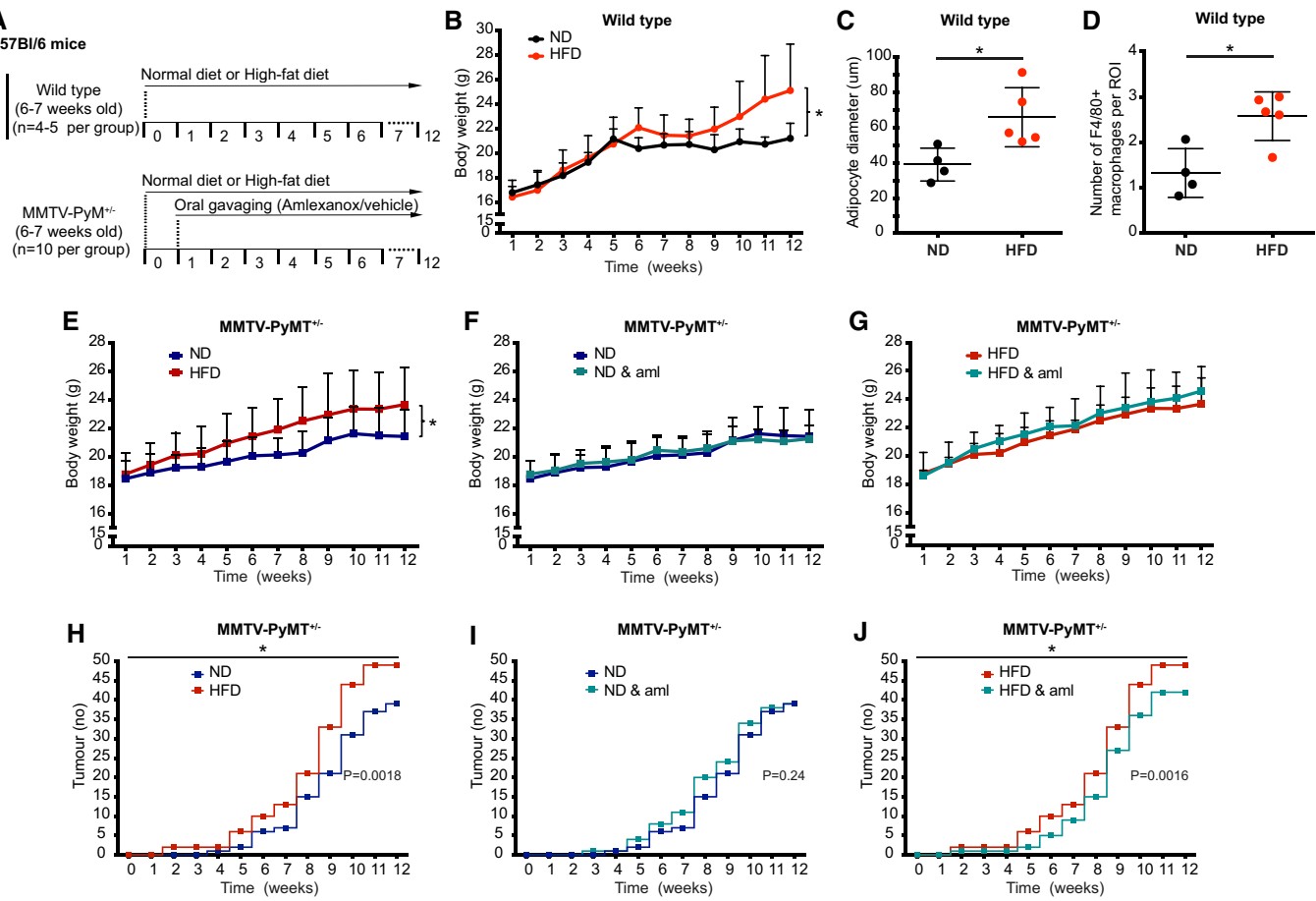

**Figure 7. Amlexanox delays tumour formation in a diet-induced obesity/PyMT-MMTV mouse model.**

Ten per group female C57Bl/6 mice heterozygous for the PyMT-MMTV started high-fat diet (HFD) when 6- to 7-week-old (prior of mammary tumour development) together with control mice on normal diet (ND; *n* = 10). Mice were administered amlexanox (17 mM) or vehicle control daily by oral gavage (Reilly *et al*, 2013). A control group (wild-type mice) was also included to confirm the efficacy of the diet and to control for possible unpredicted interactions of the MMTV-PyMT gene.

A Schematic representation of combined genetic mouse model of breast cancer and high-fat diet-induced obesity used to study the effects of amlexanox in tumour formation.

B Wild-type mice on HFD gain more weight compared to ND mice. *P < 0.05 by one-tailed Student's *t*-test.

C Average adipocytes size (major diameter) in mammary fat pads is larger in HFD mice compared to ND mice based on the analysis of at least 100 adipocytes per mice on H&E-stained sections. *P < 0.05 by two-tailed Student's *t*-test.

D The number of F4/80 + macrophages in mammary fat pads is increased in HFD mice compared to ND mice. The average of F4/80 + macrophages in at least 10 fields of view per mouse is shown. *P < 0.05 by two-tailed Student's *t*-test.

E PyMT-MMTV[+/−] mice on HFD gain more weight compared to ND mice. *P < 0.05 by one-tailed Student's *t*-test.

F, G Amlexanox does not affect weight gain either in mice on F) ND or G) HFD.

H Mammary tumours are developing earlier in PyMT-MMTV[+/−] mice on HFD compared to ND mice. *P* value was calculated using log-rank test.

I, J Amlexanox does not affect tumour development in PyMT-MMTV[+/−] mice on (I) ND, but delays tumour formation in (J) HFD mice. *P* value was calculated using log-rank test.

Data information: Lines and error bars represent mean and SD. Exact *P* values are shown in Table EV3.
Source data are available online for this figure.

decreased by HFD (Fig 7H), and while no effect of amlexanox was observed on tumour appearance in mice on ND (Fig 7I), the effect of HFD was significantly reverted by amlexanox treatment (Fig 7J).

These data showed that diet-induced obesity synergizes with MMTV-PyMT in promoting tumour formation and inhibition of IKKε/TBK1 by amlexanox delays tumour appearance in mice on HFD, even when induced by a strong oncogene, such a PyMT.

# Discussion

Obesity is estimated to be the cause of cancer in 14% of cases in men and around 20% in women (Calle *et al*, 2003). Despite many indications that a critical factor increasing the risk of cancer in obese patients is obesity-associated inflammation, partially caused by infiltrating macrophages, little is known about the mechanism(s) underlying this phenomenon. While macrophages were shown to play a

crucial role at different stages of cancer progression (Williams *et al*, 2016), here we studied their role in promoting the very early stages of malignant transformation, using macrophage-conditioned medium derived from 25 healthy donors.

We report that macrophages induce acquisition of malignant phenotypes in three independent model systems: (i) the non-tumourigenic MCF10A cell line, (ii) mouse primary organoids and (iii) human primary breast epithelial cells derived from patients. We also further show that acquisition of malignant properties induced by macrophage-conditioned medium can be prevented by genetic (via CRISPR/Cas9) inhibition of IKKε and PHGDH and pharmacologically, via amlexanox, a recently identified inhibitor of IKKε/TBK1, and NCT502, an inhibitor of PHGDH. Moreover, amlexanox delayed tumour appearance *in vivo* in a combined diet-induced obesity/MMTV-PyMT breast cancer model.

Amlexanox is an FDA-approved drug for the treatment of apthous ulcers, rediscovered as IKKε/TBK1 inhibitor, that in mice improves obesity-related metabolic dysfunction (Reilly *et al*, 2013), suggesting that it targets processes linking inflammation to metabolism. Recently, the drug has been tested in a proof-of-concept study in obese patients with type 2 diabetes and non-alcoholic fatty liver disease, where it improved metabolic parameters in a subset of patients characterized by a specific inflammatory signature in subcutaneous fat. Importantly, the safety of the use of amlexanox in humans was confirmed, with the only side effect reported being rash, in the majority of the cases classified as non-clinically relevant (Oral *et al*, 2017). The level of inflammation in the white adipose tissue is considered an important factor distinguishing metabolically unhealthy obese patients from metabolically healthy ones and is also a characteristic of 20% of the metabolically unhealthy lean population (Stefan *et al*, 2017). Thus, in the context of a potential clinical follow-up of our study, the level of inflammation in the breast of obese patients should be a key parameter to be used for stratifying patients and to increase the probability of a successful preventive effect. Importantly, in humans, as in our study, amlexanox did not cause a significant reduction in body weight, suggesting that its most likely mechanism of action is a direct preventive effect of inflammation, and not an event secondary to the effect on whole-body metabolism.

The other compound we tested, the PHGDH inhibitor NCT502, shows efficacy *in vivo* to reduce growth of tumours overexpressing PHGDH (Pacold *et al*, 2016). Here, we report a possible preventive role for this molecule in obese patients with signs of inflammation, also highlighting a new regulatory mechanism for SBP.

An unexpected finding of our study is that media conditioned by the four macrophage populations, independently of the activation state, led to acquisition of malignant characteristics. The fact that differentiated macrophages, before activation, induce the malignant phenotype can be explained by their secretory activity (Fig EV1C–J and Table EV1). Moreover, while the four different conditioned media induce a similar malignant phenotype, it is important to highlight that from a mechanistic point of view M1 and M2, macrophages use different strategies, with M1 (differentiated or activated), but not M2, being able to overcome the proliferative arrest in MCF10A spheroids. Thus, while a common factor might be secreted by the four cell types (Jablonski *et al*, 2015), another possible scenario is that the same phenotype is achieved by different signalling pathways, activated by different cytokines.

Importantly, we found that the expression of CD206 is characteristic of the M1D and M2A populations (Fig EV2E and G). Recently, macrophages present in the breast of obese women were shown to express the same marker and to resemble more M2 than M1 subtype, bearing also similarities to tumour-associated macrophages (Springer *et al*, 2019). Macrophages in the breast have also been reported to reprogramme during obesity, become metabolically activated and different from M1 macrophages due to their pro-tumourigenic activity (Tiwari *et al*, 2019). However, since the extent of macrophages around dying adipocytes (called crown-like structures) correlates with increased levels of pro-inflammatory cytokines in the circulation (Morris *et al*, 2011), these macrophages have also been considered M1. In conclusion, while there is still some controversy in the field, it is important to consider that macrophages are an heterogeneous population and M1 and M2 are oversimplified extremes. Future work will fully characterize the population of macrophages localized in the breast tissue during obesity.

Equally, the contribution of other immune cells, reported to infiltrate the breast tissue *in vivo,* remains to be elucidated, together with adipocytes and other stromal cell types, such as fibroblasts.

Finally, our study shows the feasibility of inhibiting IKKε and its downstream signalling, in particular the SBP, as shown by our recent work (https://biorxiv.org/cgi/content/short/855361v1), as preventive strategy to reduce the risk of breast cancer associated with obesity.

# Materials and Methods

### Reagents and antibodies

All reagents were obtained from Sigma-Aldrich, UK, unless stated otherwise.

### Cells

MCF10A cell line (ATCC; LGS Standards UK) was cultured in DMEM/F12 medium supplemented with 5% horse serum (HS), 20 ng/ml epidermal growth factor (EGF), 500 ng/ml hydrocortisone, 10 μg/ml insulin, 100 ng/ml cholera toxin and normocin (InvivoGen, UK). For 3D cultures, the cells were grown in the same medium with reduced HS (2%) and EGF (5 ng/ml) and with an addition of 2% Matrigel (Debnath *et al*, 2003).

### Macrophage differentiation

Peripheral blood mononuclear cells (PBMCs) were isolated from leucocyte cones of healthy deidentified blood donor volunteers from the NHS Blood and Transplant bank, and experiments were conducted in accordance with a protocol approved by Queen Mary Research Ethics Committee (QMREC 2014:61) and in accordance with the Helsinki Declaration. PBMCs were isolated following density gradient centrifugation using Histopaque 1077. These were then plated in 10-cm$^2$ plate (Greiner) at a density of $2 \times 10^7$ cells/plate, and monocytes were allowed to adhere to the culture plates for 1 h in PBS ($Ca^{2+}/Mg^{2+}$). Non-adhering cells were removed with washes with PBS $Ca^{2+}$ and $Mg^{2+}$ free. Monocyte-to-macrophage differentiation was induced for 7 days in RPMI 1640 medium

supplemented with 10% foetal bovine serum towards either M1-differentiated macrophages (M1D; 50 ng/ml granulocyte–macrophage colony-stimulating factor, GM-CSF; PeproTech) or M2-differentiated macrophage (M2D; 50 ng/ml macrophage colony-stimulating factor, M-CSF; PeproTech). For macrophage activation, M1D macrophages were stimulated with 10 ng/ml LPS (InvivoGen, UK) and 20 ng/ml IFNγ for 24 h (M1A), whereas M2D macrophages were stimulated with 20 ng/ml IL-4 for 48 h (M2A) (Martinez *et al*, 2006; Däbritz *et al*, 2015; Fig EV1A).

### Conditioned medium preparation

Activated or differentiated macrophages were rinsed 4 times with PBS and cultured in DMEM/F12 medium supplemented with 5% HS and normocin for 24 h. The same medium conditioned by MCF10A cells for 24 h was used as control medium. The conditioned medium was collected, filtered (0.45 μm) and stored at 4°C for up to 1 month.

### ELISA

TNF-α (pg/ml) and CCL22 (ng/ml) concentrations were determined in medium conditioned by either macrophages or MCF10A cells (used as control medium) with an ELISA kit according to the manufacturer's instructions (R&D Systems, UK).

### Flow cytometry

Macrophages were stained as previously described (Georgouli *et al*, 2019). Macrophages were washed three times with $PBS^{-/-}$, incubated for 5 min at 37° in $PBS^{-/-}$ and then gently scraped and washed once with FACS buffer ($PBS^{-/-}$, 1% BSA, 2 mM EDTA, 0.1% $NaN_3$). Nonspecific binding sites were blocked with Fc Receptor Blocking Solution (Human TruStain FcX; BioLegend), and then, the cells were co-stained with primary antibodies anti-HLA-DR fluorescein isothiocyanate (FITC)-conjugated (1:50; 11-9952-42; Thermo Fisher Scientific), anti-CD86 phycoerythrin–cyanine 7 (PE-Cy7) (1:20; Clone IT2.2; Thermo Fisher Scientific), anti-CD163 allophycocyanin (APC)-conjugated (1:20; 17-1639-42; Thermo Fisher Scientific) and anti-CD206 PE-conjugated (1:20; Clone 15-2; BioLegend) for 30 min at 4°C in the dark. Cells were washed twice with FACS buffer and stained with DAPI (5 μg/ml) for live/dead cell separation immediately before data acquisition on BD LSRFortessa 1 flow cytometer (BD Biosciences). The data were analysed using FlowJo 10.6.1 software (Tree Star, Inc).

### Cytokine array

Cytokines from medium conditioned either by macrophages or MCF10A cells were analysed with human cytokine antibody array (ab133997; Abcam) according to the manufacturer's instructions. Uncultured media were tested to assess baseline signal responses. Signals were detected with chemiluminescence reaction, and densitometry analysis was performed with ImageJ software (https://imagej.nih.gov/ij/).

### Proliferation assay

MCF10A cells were plated in 96-well plates (Corning) at a density 1,000 cells per well. Cell proliferation was determined by measuring cell confluence over 5 days using the IncuCyte ZOOM instrument (Essen BioScience, Ann Arbor, MI, USA) and analysed with the IncuCyte ZOOM 2015A software.

### Wound-healing assay

MCF10A cells were plated in 96-well plates (IncuCyte® ImageLock Plates; Essen BioScience, UK) at a density 50,000 cells per well and grown ON until confluent. To block proliferation, the cells were treated with mitomycin C (10 μg/ml) for 2 h and wounds were made with a 96-pin WoundMaker (Essen BioScience). Cell migration was determined by measuring cell confluency within the wound region over 2 days using the IncuCyte ZOOM instrument (Essen BioScience, Ann Arbor, MI, USA) and analysed with the IncuCyte ZOOM 2015A software.

### CRISPR cell line generation

CRISPR-Cas9 system was used to target either *IKBKE* (crRNA-495656, Dharmacon™; crRNA IKKε) or *PHGDH* gene (crRNA-497793; Dharmacon™; crRNA PHGDH) in MCF10A cells stably expressing Cas9 (a generous gift from Hyojin Kim, FeiFei Song and Chris Lord ICR—London, UK). MCF10A-Cas9 cells were seeded at a density of $1.5 \times 10^5$ cells per 6-cm-diameter well (Corning). On the following day, cells were transfected with crRNA IKKε, crRNA PHGDH or control non-targeting crRNA. The crRNA was mixed with tracrRNA in a 1:1 ratio (2 μM) and then incubated with Opti-MEM (Thermo Fisher Scientific) in a 1:3 ratio for 5 min at RT. RNaiMAX Lipofectamine (Thermo Fisher Scientific)/Opti-MEM mix (1:19 ratio) was prepared independently. CrRNA:tracrRNA mix was added dropwise to Lipofectamine/Opti-MEM mix for a final concentration of crRNA:tracrRNA of 50 nM and left for 20 min at RT. Transfection mix was added dropwise to the MCF10A-Cas9 cells and incubated for 6 h at 37°C. After that, the cells were rinsed with PBS and further cultured for 4 days in the growth medium. Clonal selection was performed by plating single cells in individual wells of a 96-well plate (Corning). Growing colonies were expanded to 12-well plate format and checked for either IKKε or PHGDH expression by Western blotting. Cells with either IKKε or PHGDH knockout or cultures derived from single clones of MCF10A-Cas9 cells treated with control non-targeting crRNA were used for further analyses. To avoid confounding effects of single-cell clones, we combined together 3 clones for each condition (control, IKKε and PHGDH) that were further used as pools.

### Western blotting

For Western blot analyses, cells were lysed in a cold lysis buffer (20 mM Tris–HCl, pH 7.4, 135 mM NaCl, 1.5 mM $MgCl_2$, 1% Triton, 10% glycerol) containing protease (Roche, UK) and phosphatase (Thermo Fisher Scientific) inhibitor cocktails. Protein content was quantified, and equal amounts (20 μg) prepared in NuPAGE LDS sample buffer (Thermo Fisher Scientific) were separated by SDS–PAGE using 4–12% NuPAGE™ Bis–Tris Protein gels (Thermo Fisher Scientific). After that, proteins were transferred to Immobilon-P PVDF 0.45 μm membrane (Merck). Protein was detected using primary antibodies anti-IKKε (1:1,000; 14907; Sigma), anti-TBK1 (1:1,000; 3013; Cell Signaling), anti-PHGDH

(1:4,000; HPA021241; Sigma) or anti-actin (1:2,000; sc-1615; Santa Cruz). Enhanced chemiluminescence (Thermo Fisher Scientific) was used for signal detection.

## 3D invasion assay

MCF10A cells were plated in Matrigel:Collagen I to model the extracellular microenvironment in which Matrigel represents the basement membrane, whereas collagen models the stromal extracellular matrix (Nguyen-Ngoc et al, 2012). 3D culture was assayed as previously described (Arnandis & Godinho, 2015). MCF10A cells were plated in 8-well chamber slides (BD Biosciences) at a density of 10,000 cells per chamber. For IKKε or PHGDH CRISPR-Cas9 knockout or CRISPR-Cas9 control MCF10A cells, individual clones ($n = 3$) were combined in equal ratio prior seeding in the gels. Growth factor-reduced Matrigel was used (BD Biosciences) with a protein concentration between 9 and 11 mg/ml. Collagen (Corning) was used at the concentration 1.2 mg/ml. MCF10A cells were cultured in the assay medium with reduced HS (2%) and EGF (5 ng/ml) and 2% Matrigel (Godinho et al, 2014) with medium replaced every 4 days. On day 16, MCF10A spheroids were stimulated with either macrophage-conditioned or control medium supplemented with 20 ng/ml EGF, 500 ng/ml hydrocortisone and 10 μg/ml insulin. For IKKε inhibition, MCF10A spheroids were stimulated with conditioned medium in the presence of specific inhibitor of IKKε/TBK1-amlexanox (50 μM; Abcam) (Reilly et al, 2013). To inhibit the serine biosynthesis pathway, the PHGDH inhibitor NCT502 (10 μM; Cayman) was used (Pacold et al, 2016). For Rac1 inhibition, the spheroids were treated with NSC23766 (50 μM; EMD Millipore) (Godinho et al, 2014). After 24 h, invasion was quantified microscopically by visual scoring.

## Immunofluorescence microscopy on 3D cultures

MCF10A spheroids were fixed and stained as previously described (Arnandis & Godinho, 2015). Spheroids were washed in PBS and fixed with 4% PFA (Thermo Fisher Scientific) in PBS for 20 min at 37°C. Then, the spheroids were rinsed with PBS:glycine (100 mM) in PBS 3 times for 10 min each and permeabilized with 0.5% Triton X-100 for 10 min. Permeabilized spheroids were rinsed for 10 min with 10% FBS in immunofluorescence buffer (IF; 130 mM NaCl, 13 mM $Na_2HPO_4$, 3.5 mM $NaH_2PO_4$, 7.7 mM $NaN_3$, 0.1% bovine serum albumin, 0.2% Triton X-100, 0.05% Tween-20 at pH 7.4) and blocked with 10% FBS-IF buffer for 1 h. Then, the spheroids were incubated with primary antibodies anti-Laminin V Alexa 488-conjugated (1:200; Millipore) or anti-Ki67 Alexa 488-conjugated (1:100, BD Pharmingen) in 10% FBS-IF buffer ON at 4°C. Spheroids were washed with IF buffer three times for 20 min each and then incubated with Alexa Fluor® 633 phalloidin for β-actin staining (1:500; Thermo Fisher Scientific) for 1 h in 10% FBS-IF buffer. The spheroids were rinsed 3 times for 10 min and stained for DNA with Hoechst 33342 (1:2,500, BD Pharmingen) for 5 min, and washed with PBS for 10 min. 3D cultures were mounted in ProLong Gold Antifade mounting medium (Thermo Fisher Scientific). The spheroids were analysed under Zeiss LSM 710 confocal microscope (Carl Zeiss Microscopy, U.S.) using Zen 2009 software. For quantification of lumen filling, at least 50 spheroids in duplicate were scored per condition for each experiment according to the criteria for lumen filling with cell nuclei: filled (~90–100% filled), almost filled (~50–90% filled), partially filled (~10–50% filled) and clear (~0–10% filled) as shown in Fig 1B. For Ki67 staining analysis, the percentage of Ki67-positive nuclei/spheroid was determined.

## Soft agar assay

MCF10A, IKKε or PHGDH CRISPR-Cas9 knockout MCF10A cells, or CRISPR-Cas9 MCF10A control cells were seeded in 0.4% soft agar in macrophage-conditioned or control medium on a layer of 0.6% soft agar in the same medium in 24-well plate (Corning) at a density of 5,000 cells per well ($n = 3$ per condition). For IKKε or PHGDH CRISPR-Cas9 knockout or CRISPR-Cas9 control cells, individual clones ($n = 3$) were combined in equal ratio prior seeding in soft agar. Cultures were fed every 7 days with 0.4% soft agar in the same medium. For IKKε inhibition, amlexanox (50 μM) was added to the medium. To inhibit serine biosynthesis pathways, NCT502 (10 μM) was added to the medium. After 5 weeks, colonies greater than 50 μm in diameter were quantified microscopically.

## Organoids

Mammary organoids from 19- to 20-week-old C57Bl/6 mice that were either on high-fat diet (HFD; body weight: 27.2 g ± 1.48) or normal diet (ND; body weight: 21.3 g ± 1.27) were isolated and plated for 3D cultures as previously described (Nguyen-Ngoc et al, 2015) ($n = 3$ per group). Briefly, fat pads were minced and then enzymatically digested in collagenase solution in DMEM/F12 medium (2 mg/ml collagenase, 2 mg/ml trypsin, 5% FBS, 5 μg/ml insulin and 50 μg/ml gentamicin) for 40–50 min at 37°C on a shaker (130 rpm). Organoid suspension was treated with 2 U/μl DNase for 5 min to detach organoids from single cells, and organoids were purified from stromal cell populations by differential centrifugation (176 g for 3–4 s, × 4 spins). Isolated organoids were assayed in two different culture set-ups of extracellular matrix and were plated either in Matrigel:Collagen I or Collagen I in 8-well chamber slides (BD Biosciences).

## Organoids: Invasion assay in Collagen/Matrigel

Organoids were plated at a density of 1.8 organoids per 1 μl of Matrigel:Collagen I. Collagen was used at the concentration 2.4 mg/ml. Organoids were cultured in DMEM/F12 medium supplemented with 1% insulin–transferrin–selenium–X (ITS, Gibco), 2.5 nM fibroblast growth factor 2 (FGF2) and 1% penicillin/streptomycin. After 2 days, the organoids were stimulated with either macrophage-conditioned or control medium supplemented with 1% ITS and 2.5 nM FGF2. Amlexanox (50 μM) was used to inhibit IKKε (Reilly et al, 2013). NCT502 (10 μM) was used to inhibit PHGDH (Pacold et al, 2016). After 24 h, invasion was quantified microscopically.

## Organoids: Invasion assay in collagen

Previous studies reported that collagen I induces in both normal and tumour organoids, a conserved response of protrusive invasion (Nguyen-Ngoc et al, 2012). To test possible differences in responses of protrusive invasion between HFD and ND organoids, organoids

were assayed in Collagen I. Organoids were plated at a density of 1.8 organoids per 1 µl of Collagen I (3 mg/ml) and cultured in DMEM/F12 medium supplemented with 1% ITS, 2.5 nM FGF2 and 1% penicillin/streptomycin ON. Amlexanox (50 µM) or NCT502 (10 µM) was added to the culture medium for further 24 h. Then, the number of invasive protrusions per each organoid was quantified microscopically with 30 organoids scored per condition.

### Organoids: Immunofluorescence microscopy

Organoids were fixed and stained as previously described (Nguyen-Ngoc et al, 2015). Briefly, organoids were washed in PBS and fixed with 4% PFA in PBS for 15 min at RT. Then, the organoids were washed in PBS 3 times for 10 min each and permeabilized with 0.5% Triton X-100 for 40 min. Permeabilized organoids were blocked with 10% FBS in PBS for 2 h and then incubated with anti-α-SMA antibody (1:200) in 10% FBS-PBS ON at 4°C. Organoids were washed with 10% FBS-PBS 3 times for 10 min each and incubated with Alexa 488-conjugated secondary antibody (1:500; Thermo Fisher Scientific) and Alexa Fluor® 633 phalloidin (1:500) for 1 h in 10% FBS-PBS. The spheroids were washed in PBS 3 times for 10 min and incubated with Hoechst 33342 (1:2,500) for 5 min, and washed with PBS for 10 min.

3D cultures were mounted in ProLong Gold Antifade mounting medium (Thermo Fisher Scientific). The spheroids were analysed under Zeiss LSM 710 confocal microscope (Carl Zeiss Microscopy, U.S.) using Zen 2009 software.

### 3D ductal culture

Myoepithelial and luminal cells isolated from human breast tissue were previously shown to reform ductal structures with internal luminal cell layer and external myoepithelial cell layer, and with luminal compartment when grown in collagen gels (Carter et al, 2017). Furthermore, in the same model, inducible expression of HER2 in luminal compartment was shown to induce luminal filling, recapitulating ductal carcinoma in situ. Thus, this 3D model appears to recreate physiologically reflective duct (Carter et al, 2017).

Pure populations of myoepithelial and luminal cells (n = 2) were obtained from the Breast Cancer Now Tissue Bank at the Barts Cancer Institute (REC:15/EE/0192). All patients donated tissues from which cells were derived following fully informed consent, and all experiments conformed to the principles set out in the WMA Declaration of Helsinki and the Department of Health and Human Services Belmont Report. Luminal cells were cultured in DMEM/F12 medium supplemented with 10% FBS, 0.5 µg/ml hydrocortisone, 10 µg/ml apo-transferrin, 5 µg/ml insulin and 10 ng/ml EGF. Myoepithelial cells were cultured in HuMEC medium (Thermo Fisher Scientific) supplemented with 0.5 µg/ml hydrocortisone, 5 µg/ml insulin, 10 ng/ml EGF and 50 µg/ml bovine pituitary extract (Thermo Fisher Scientific; Carter et al, 2017).

### 3D ducts: lumen filling assay

Myoepithelial and luminal cells were combined in a 1:1 ratio $(1 \times 10^4$ cells each) and seeded into 2 mg/ml collagen I type gels as previously described (Carter et al, 2017). Cells were grown for 14 days to reform ductal structures in luminal culture medium with

medium replaced every 2-3 days. From day 14, 3D ducts were cultured in macrophage (M1A or M2A)- or MCF10A-conditioned medium supplemented as the luminal culture medium. Amlexanox (50 µM) was used to inhibit IKKε (Reilly et al, 2013). NCT502 (10 µM) was used to inhibit PHGDH (Pacold et al, 2016).

### 3D ducts: Immunofluorescence microscopy

3D ducts were fixed and stained as previously described (Carter et al, 2017). Briefly, the collagen gels were treated with 1 mg/ml collagenase, and then, the ducts were fixed with 10% neutral-buffered formalin for 10 min at 37°C and then permeabilized ON with 1% Triton X-100 and blocked in 10% FBS/2%BSA. The gels were incubated with 1 µM Alexa Fluor® 564 phalloidin for β-actin staining (A22283, Thermo Fisher Scientific) for 10 min and stained for DNA with 1 µg/ml DAPI prior to mounting.

Fluorescent images were acquired using Zeiss LSM 710 confocal microscope (Carl Zeiss Microscopy, U.S.). For quantification of lumen filling, 3 ducts in duplicate were scored per each condition according to previously described eight-step image analysis process (Carter et al, 2017). For each 3D duct, raw DAPI-labelled z-sections were analysed to calculate the total area of the sphere and the area of luminal space in the centre. Filling of the lumen was determined as % of luminal space filled with cells.

### Mouse models

Female C57Bl/6 mice heterozygous for the polyomavirus middle T antigen (PyMT) under the control of the mouse mammary tumour virus promoter (MMTV) started high-fat diet (HFD; 45% of calories come from fat: TestDiet) at 6–7 weeks of age, in parallel to mice on normal diet (ND; 15% of calories come from fat: LabDiet®). A week after starting the diet, the mice were administered amlexanox (25 mg per kg body weight, 17 mM) (Reilly et al, 2013) or vehicle control (250 mM Tris, 45 mM NaOH, pH 7.5) daily by oral gavage. The appearance of new tumours was monitored twice per week. A control diet group (wild-type mice) was included to confirm the efficacy of the diet and to control for possible unpredicted interactions of the MMTV-PyMT gene. The body weight of the mice was measured once per week.

To quantify adipocyte diameter, mammary fat pads were fixed in 10% formalin, paraffin-embedded, sectioned (4 µm), fixed on slides and H&E-stained. Average adipocyte diameter was calculated by measuring the major diameter of at least 100 adipocytes per mice using AxioVision software 4.8 (Zeiss).

To quantify macrophage infiltration in mammary fat pads, tissue sections fixed on slides were processed for immunohistochemistry (IHC) using standard IHC procedures. Briefly, antigen retrieval was performed by enzymatic digestion using protease 1 (Roche, UK), tissue sections were stained for a macrophage marker with rat anti-F4/80 antibody (1:300; Bio-Rad, UK), and the staining was visualized with DAB (Vector Laboratories, UK) reaction. The number of macrophages was determined by counting F4/80 + macrophages in at least 10 fields of view (depending on tissue availability) and averaged per each mouse.

All animal procedures were approved by the animal ethics committee of Queen Mary of London and were performed in accordance with United Kingdom Home Office regulations.

## Immunohistochemistry

Breast tissue samples from 66 female patients diagnosed with primary breast carcinoma between the period 2013 and 2016 were obtained from the BCI Breast Tissue Bank (Ethics Ref: 15/EE/0192). All patients donated tissue following fully informed consent, and all experiments conformed to the principles set out in the WMA Declaration of Helsinki and the Department of Health and Human Services Belmont Report. The patients were of mixed race and ethnicity with the white British as a largest group (38%). The mean age of the patients was 60.6 years ± 12.8 SD, and 79% of patients were post-menopause.

Tumours were characterized by invasive tumour grade and size. According to the invasive grade status, 56% of tumours were classified as grade 3 tumours, 35% as grade 2 and 9% as grade 1. Average tumour size was 24 mm ± 14.8 SD. To determine IKKε expression, tumours were fixed in 10% formalin, paraffin-embedded, sectioned (4 μm) and fixed on slides. Tissue sections were processed for IHC using standard IHC procedures. Briefly, antigen retrieval was performed by boiling tissue sections in microwave in antigen unmasking solution, citric acid-based (Vector Laboratories, UK). Tissue sections were stained for IKKε with rabbit anti-IKKε C-terminal (1:400; SAB1306435; Sigma), and the staining was visualized with DAB (Vector Laboratories, UK) reaction. To determine PSAT1 expression ($n = 41$), tissue sections were processed for IHC using Ventana Discovery®XT automated slide staining system (Roche, UK). Tissue sections were stained for PSAT1 with rabbit anti-PSAT1 (1:50; 20180-1-AP; Proteintech, UK), and the staining was visualized with DAB reaction (Roche, UK). Protein expression was assessed semi-quantitatively using a 0-3 grade scale ("0": no staining; "1": weak staining; "2": moderate staining; "3": strong staining).

The specificity of IKKε antibody for IHC analysis was determined by confirming a positive staining in IKKε expressing breast cancer cell line MDA-MB-468 and the lack of IKKε staining in 293-Flp-In cells that do not express the kinase by IHC. For IHC analysis of PSAT1, the specificity of the PSAT1 antibody was determined in MDA-MB-453 breast cancer cell line with doxycycline-inducible PSAT1 gene.

Immune cell infiltration in the breast tissue was assessed semi-quantitatively by histopathology examination in the sections previously stained by IHC for IKKε using (0–4) inflammation grade ("0": no inflammatory cells; "1": weak inflammation; "2": moderate inflammation; "3": strong inflammation; "4": very strong inflammation).

## Gene expression analysis

The METABRIC gene expression dataset (Curtis *et al*, 2012) was obtained from Synapse: http://www.synapse.org (syn1688369/METABRIC Data for Use in Independent Research). ESTIMATE analysis to predict the presence of stromal/immune cells in tumour tissues was performed using the ESTIMATE R-package version 1.0.13 (http://r-forge.r-project.org/R/?group_id = 2237). ESTIMATE algorithm is based on single-sample Gene Set Enrichment Analysis and generates three scores: (1) stromal score (that captures the presence of stroma in tumour tissue), (2) immune score (that represents the infiltration of immune cells in tumour tissue) and (3) estimate score (that infers tumour purity; Yoshihara *et al*, 2013). Immune scores for each sample were calculated, and correlation to $\log_2$-normalized gene expression was computed using Pearson's method. Correlation values and adjusted r-squared are shown.

## Statistics

Statistical analysis was performed using GraphPad Prism 8.0 or R software (version 3.5.0). For parametric data analyses, Student's *t*-test was used for comparison between two groups or one-way ANOVA with uncorrected Fisher's LSD *post hoc* test when comparing more than two groups. For *in vivo* data, to determine the statistical significance between the numbers of developed tumours across all time points, log-rank test was used. To evaluate IKKε expression in breast carcinomas, Kruskal–Wallis test with uncorrected Dunn's *post hoc* test was used. The associations between protein expression and immune cell infiltration in breast carcinomas were assessed using Spearman's correlation. $P < 0.05$ was considered statistically significant.

**Expanded View** for this article is available online.

## Acknowledgments

The authors wish to acknowledge the role of the Breast Cancer Now Tissue Bank in collecting and making available the tissue and primary cell samples used in the generation of this publication, the Barts Cancer Institute pathology facility and George Elia for help and support during this study, Elzbieta Stankiewicz for help with immunohistochemical staining procedure, Teresa Arnandis for guidance on 3D assays and Tencho Teven, Hyojin Kim, FeiFei Song and Chris Lord for guidance in the generation of the MCF10A-CRISPR cells. We are grateful to Julie Cleaver and Tracy Chaplin-Perkins from the Animal Technical Service team for helping with mice care and oral gavage during the *in vivo* study. We thank Shannon Reilly and Alan Saltiel for useful guidance on the use of amlexanox *in vivo* and Jacek Marzec for helping with statistical analysis. We are also grateful to Mirella Georgouli, Victoria Sanz-Moreno and Agata Krygowska for their guidance in the characterization of macrophages at the FACS. We would like to thank Robert James Hearnden for feedback on this manuscript. KB and EW-V are supported by the Barts London Charity (Grant Reference Number: 467/2053). GS is funded by University College London COMPLeX/British Heart Foundation Fund (SP/08/004), the BBSRC (BB/L020874/1) and the Wellcome Trust (097815/Z/11/Z) in the UK, and the Italian Association for Cancer Research (AIRC, IG13447). JD is supported by a Sir Henry Dale Fellowship jointly funded by the Wellcome Trust and the Royal Society (grant no. 107613/Z/15/Z). EC and RG are supported by Breast Cancer Now (Grant: 2017NovPR988), and KH-D is funded by CRUK (programme grant C8218/A21453).

## Author contributions

EW-V designed and performed the majority of the experiments described in the manuscript. EC and RG conducted and analysed experiment on 3D human duct model. AI analysed mouse tissues. RX performed WB and helped with clonal selection. IM conducted transfection. JH and KH-D helped with the planning and execution of the *in vivo* experiments. WJ helped with optimization of PSAT1 staining in human tissue. RMB and LU helped with the analysis of mouse tissues. SAG provided support and guidance in establishing the 3D model system with MCF10A and organoids. JD provided support in differentiating macrophages from PBMCs. GS and RBB analysed the METABRIC dataset. LJ analysed and quantified the tissue sections of human breast cancer cases. KB designed the study and wrote the manuscript with the help of EW-V, GS, SAG, KH-D, JD and LJ.

**The paper explained**

**Problem**

Obesity is constantly increasing in developing countries. One in 20 cases of cancer can be attributed to obesity, which is now considered the second most preventable cause of cancer after smoking. In particular, the low level of chronic inflammation associated with obesity is a key factor responsible for the higher risk of cancer development, also in the case of breast tumours, where macrophages infiltrating the breast tissue give rise to an inflammatory response. However, the exact role of macrophages in promoting tumourigenesis remains elusive as well as the mechanism by which they act. Thus, the purpose of our study was to explore the very early stages of breast cancer during obesity and formally investigate the mechanisms by which macrophages promote tumourigenesis.

**Results**

Our data demonstrate that medium conditioned by macrophages derived from human healthy donors induces acquisition of a malignant phenotype in different 3D systems: (i) the non-transformed MCF10A cells, (ii) mouse primary organoids and (iii) human primary breast epithelial cells derived from patients. This transforming effect of macrophages is prevented by the FDA-approved drug amlexanox, inhibitor of the innate immunity kinases IKKε/TBK1 and also by genetic inhibition of IKKε (via CRISPR/Cas9). Furthermore, in line with the role of IKKε as regulator of the serine biosynthesis pathway, we show that the inhibition of this pathway via the newly developed compound NCT502 (targeting PHGDH) prevents macrophage-induced transformation as well as deletion of *PHGDH* gene via CRISPR/Cas9. Amlexanox delays tumour appearance in a combined genetic mouse model of breast cancer and high-fat diet-induced obesity/inflammation *in vivo*. We also validated the link between inflammation–IKKε and alteration of cellular metabolism further in translational studies. We show a positive correlation between IKKε expression, immune cell infiltration and the expression of the serine biosynthesis pathway enzyme PSAT1 in human breast cancer tissues and in the METABRIC breast cancer dataset.

**Impact**

Here, we describe a mechanism linking macrophage-mediated activation of the innate immune system to the very early stages of malignant transformation of breast epithelial cells. We show that macrophages derived from human healthy donors promote malignant transformation. Moreover, inhibition of either the innate immune kinases IKKε/TBK1 or the serine biosynthesis pathway prevents the phenotype. Since amlexanox safety has been confirmed in human patients, we propose that this drug is an interesting candidate to be tested in preventive therapeutic strategies, aiming to reduce the risk of breast cancer associated with obesity.

## Conflict of interest

The authors declare that they have no conflict of interest.

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
