## [Review Process File · EMBO Molecular Medicine]

Macrophages induce malignant traits in mammary epithelium via IKK ϵ /TBK1 kinases and the serine biosynthesis pathway

Ewa Wilcz-Villega, Edward Carter, Alastair Ironside, Ruoyan Xu, Isabella Mataloni, Julie Holdsworth, William Jones, Rocio Moreno Bejar, Lukas Uhlik, Robert B. Bentham, Susana A. Godinho, Jesmond Dalli, Richard Grose, Gyorgy Szabadkai, Louise Jones, Kairbaan Hodivala-Dilke & Katiuscia Bianchi

Review timeline:

Submission date:	21 February 2019
Editorial Decision:	5 April 2019
Revision received:	16 October 2019
Editorial Decision:	11 November 2019
Revision received:	3 December 2019
Accepted:	4 December 2019

Editor: Lise Roth

Transaction Report:

1st Editorial Decision

5 April 2019

Thank you for the submission of your manuscript to EMBO Molecular Medicine. We have now heard back from the three referees who were asked to evaluate your manuscript.

As you will see from the reports below, they all acknowledge the potential interest of the findings for the field, but they also have fundamental and overlapping concerns that should be addressed in a major round of revision of the present manuscript, so that the data fully support the conclusions.

Considering that at EMBO Press we encourage one round only of major revisions within a 3 months timeframe, a cross-commenting exercise helped defining the main points that should be addressed. Therefore, in addition to the several editorial changes requested, a minimum set of experiments should include:

- 1) At least the key experiments performed with a second cell line, or even better with primary cells.
- 2) At least the key experiments performed with genetic inhibition, to strengthen pharmacological inhibition.
- 3) A careful characterization of macrophage polarization and secreted cytokines/chemokines.
- 4) A careful comparison of statistically significant differences among the groups where indicated.
- 5) Improving in vivo experiments must at least be attempted: the referees recognize that a second transgenic model may take too long to establish, but a xenograft approach could be used (better in a syngeneic mouse model).
- 6) The first in vivo experiment should be removed (point 13 from referee #3).

Addressing these points will be necessary for further considering the manuscript in our journal. EMBO Molecular Medicine encourages a single round of revision only and therefore, acceptance or rejection of the manuscript will depend on the completeness of your responses included in the next, final version of the manuscript.

EMBO Molecular Medicine has a "scooping protection" policy, whereby similar findings that are published by others during review or revision are not a criterion for rejection. Should you decide to

submit a revised version, I do ask that you get in touch after three months if you have not completed it, to update us on the status.

Please also contact us as soon as possible if similar work is published elsewhere. If other work is published, we may not be able to extend the revision period beyond three months.

I look forward to receiving your revised manuscript.

***** Reviewer's comments *****

Referee #1 (Remarks for Author):

This report addresses an important topic, the role of breast adipose-infiltrating macrophages in breast cancer. There is strong population evidence and supporting animal model evidence that pro-inflammatory macrophages associated with insulin resistant obesity are not merely bystanders or passengers to the deterioration of metabolism and increase in cancer risk in humans with obesity, but play a causal role. Therefore deeper understanding of the relevant signaling pathways and cellular subtypes might reveal novel targets to prevent breast carcinogenesis or progression/metastasis. On the other hand, metabolism and inflammation are closely intertwined in insulin resistant obesity, such that one does not change without an effect on the other, which creates difficulties for targeting, and suggests that both metabolic and inflammatory variables should be measured in concert.

The investigators pursue new targets, non-canonical members of the IKK-family, I-kappa-B kinase-epsilon (IKK-epsilon) and TANK binding kinase 1 (TBK1) based on previously published work, and hypothesize a role in malignant transformation of breast epithelia. Overall, this manuscript reports interesting results on the effect of macrophages on the breast epithelial cells transformation, and macrophage conditioned media induce breast tumorigenesis via the kinases IKK-epsilon/TBK1 and serine metabolism. This study is significant; however some issues should be addressed.

In Figure 1, the investigators differentiate monocytes derived from PBMCs into 'M1' or 'M2' macrophages based on GM-CSF or M-CSF culture, and activation with LPS or IL-4, respectively. Positive control data to establish macrophage polarization is necessary to confirm the phenotype and should have been performed, probably by PCR array. Furthermore, the investigators surely appreciate that these two states are not polar opposites, but exist along a transcriptional spectrum and include significant individual-level differences. Finally, it is not clear what sort of polarization actually happens in human adipose tissue, where these recombinant cytokines and mitogens are not artificially introduced. These limitations to the assay and interpretation should be discussed.

How did the authors end up on IKK-epsilon/TBK1 as the linking markers of obesity, inflammation and breast tumorigenesis?

It is mentioned that MCF10A cells "are considered as a physiologically more appropriate model to monitor alterations associated with different stages of tumorigenesis". However, extrapolating the stages of tumorigenesis according to one cell line may not be conclusive.

On page 8 of the manuscript, it is claimed that IKK-epsilon mediates the macrophage-conditioned medium induced malignant phenotype, and amlexanox 50 μ M caused a strong inhibition of all cellular transformation related features in macrophage-conditioned media. However, based on Fig 4, amlexanox had similar inhibitory effect in the control samples as well. (Number of colonies (Fig 4-A), number of spheroids with invasive protrusions (4-C, 4-D and 4-E)).

The results of this paper show that PHGDH "prevents macrophage-conditioned medium induced malignant phenotype". In order to verify these results, PHGDH depleted MCF10 or MCF7 cells should be tested with macrophage-conditioned medium.

In general throughout the manuscript, the authors need to use more care when they discuss 'malignant transformation' of MCF10A cells. The phenotypes they are observing with this immortalized cell line show features (such as anchorage-independent growth) that overlap with or are consistent with 'malignant transformation' yet may not actually be truly 'malignant transformation'. Transformed cells generally do not revert to a non-transformed state, and removal of the cytokine/metabolite components that produced the phenotype switch would confirm that the effect was not 'transformation', by reversal of the phenotype. These essential control experiments appear not to have been done.

The discussion should more deeply consider the result that both 'M1' and 'M2' polarized macrophages promote aggressive phenotypes in the models. This result is unexpected, because pro-inflammatory cytokines and chemokines have been linked to tumor invasiveness, whereas immunological tolerance is linked to IL-4, Th2 polarization of T cells and M2 polarization of macrophages. The limitations to the interpretive power of the experiments should be addressed because the paradox here is not really resolved. Does the animal model show changes in infiltrating regulatory T cells for example?

Minor points. The English is not idiomatic in several places. For example, "...20% of the lean population metabolically unhealthy" (pg 13). Please have a native speaker of English proofread the manuscript before resubmission.

Referee #2 (Comments on Novelty/Model System for Author):

There are several main concerns:

- The use of only one cell line for in vitro experiments
- The use of pharmacological inhibitors to identify targets and no genetic tools.
- Obesity in the mice is modest (+ 10-20%) and this may confound some results

Referee #2 (Remarks for Author):

Obesity is a risk factor for increased incidence of estrogen receptor (ER) positive breast cancer in postmenopausal women and, to a lesser degree, of triple negative breast cancer (TMNBC) in premenopausal women. In this manuscript, Ewa Wilcz-Villega et al., investigate the role of macrophages in promoting breast cancer in obese patients. They show that breast epithelial exposed to conditioned medium of M1 and M2 differentiated and activates macrophages isolated from healthy donors, acquire malignant traits in vitro. These effects were prevented by inhibiting IKKepsilon and its homologous TBK1 with the drug amlexanox, targeting the innate immune kinase IKKepsilon and by inhibiting the phosphoglycerate dehydrogenase (PHGDH) with the drug NTC502. Amlexanox also delayed in vivo tumor formation in a genetic mouse fed with high fat diet. They used human breast cancer tissues to validate the link between inflammation, IKKepsilon and altered cellular metabolism.

Overall this is an interesting study proposing a pathway connecting obesity-driven inflammation to increased breast cancer risk and identify a potential therapeutic strategy to reverse (or contain) this risk.

There are however, several critical point and limitation that weaken the work and the conclusions.

Major comments

1. As source of macrophages the authors used blood monocytes derived from healthy donors, and differentiate in vitro toward M1 and M2 phenotype {plus minus} further activation. All populations (albeit with some differences) induced some malignant traits in MCF10A cells in vitro (multilayer spheroids, increased proliferation, invasion, lamin 5 degradation). This may be an over physiological stimulation giving rise to inflammatory stimuli way above the low-level of chronic inflammation normally associated in fat tissue with obesity. The authors provide no evidence about how representative these differentiated/activated monocytes are relative to macrophages present in the fat tissue. Can macrophages in the fat tissue reproduce these effects? This could be tested using fat tissue obtained from mammaplasty or macrophages from obese mice.

2. A second important point is that for most of the experiments *in vivo*, the authors used only MCF10A, a breast epithelia cell line known to carry tumorigenic genetic alterations, and thus more likely to respond to inflammatory stimuli compared to a normal epithelial cell. While the choice of MCF10A is understandable, a second cells line should be tested, for example murine NMuMG cells, primary human mammary epithelial cells (HMEC/HpMEC), or immortalized HMEC. There were no considerations to this regard.
3. The molecules of the proposed pathway (RAC1; PHGDH; IKKepsilon; TBK1) were functionally implicated based on the use of pharmacological inhibitors (though evidence is provided for a correlation with inflammation), only one inhibitor per target was used, and the concentrations used were rather high (up to 50 μ M, *in vitro*). In order to independently validate that identity of the targets and their active role in the process, genetic experiments are necessary (e.g. siRNA, shRNA, gene KO by CRISPR/Cas9).
4. In the MCF10A spheroid model, effects (lumen filling, invasion, Lamin degradation) appear within 24 h of exposure, which is a rather short time of transformation event to take place. Can the authors exclude that these effects are not due to increased cellular motility due to inflammation rather than to transformation events? Do MCF10A cells exposed to activated macrophages become tumorigenic *in vivo*?
5. The organoid model is interesting. Organoids derived from HFD mice were more invasive but did not (or at least it is not appreciable) filling the acini as observed in the MCF10A model. How can this difference be explained? Was there also increased proliferation?
6. In the *in vivo* experiment's obesity is rather modest (+ ca 10-20%) compared to other reports in the literature (e.g. Quail et al., DOI: 10.1038/ncb3578, + ca 50%). Can this difference be considered as surrogate of human obesity? Why larger BW differences were not attempted or achieved? This small difference may also explain the rather modest effect of HFD on tumor incidence in the MMTV model.
7. Expression of IKKepsilon, PHGDH, PAST1 was correlated with inflammation score or immune score, but there was no link with obesity. This would strengthen the analysis. Could this be performed using these samples (or other)?

Minor comments.

1. For the three pharmacological inhibitors used (RAC1/NCS23766; PHGDH/NTC502; IKKepsilon - TBK1/Amlexanox) there is no evidence of activity demonstrated in the experiments. Did the drugs used under the experimental conditions indeed inhibit those targets?
2. As stated by the authors both M1 and M2 polarized macrophages promote signs or transformation in MCF10A cells. As inflammatory macrophages in fat tissues or obese individuals are rather M1 polarized this observation is surprising and somehow confusing. Does this imply that any type of inflammation promotes transformation? Where is then the link to obesity?
3. Did the authors check whether some of the effects of macrophages conditioned medium were due (or not) to EMT? Was there any effect on cell polarity?
4. In experiment of Figure 7E, the difference of the average BW is significant but small (from ca 21 to ca 23 grams). How does this BW increase relate to human obesity? Also, in the same experiments there is quite some variability in the BW of HFD treated mice. Is there a correlation between variability in BW, inflammation in the fat tissue, and time to tumor incidence?
5. The MMTV-PyMT \pm model could be used to validate *in vitro* observations: were tumors in HFD mice more invasive, more metastatic and was this reversed by Amlexanox treatment? What about the lumen of the acini?

Referee #3 (Comments on Novelty/Model System for Author):

The MS proposes an interesting correlation between obesity, macrophage secreted factors, IKBe expression, the serine synthesis pathway and mammary tumorigenesis. The experiments are original and sound, and mostly support the conclusions. In particular, the comparison of a good number (16) of individuals from whom M1 and M2 macrophages are derived and used to generate and test conditioned media lends solidity to the study. I have detailed suggestions for improvements in the text for the authors. However, here I would like to underline that I think the *in vivo* experiments should be repeated with a less aggressive model of breast cancer.

Referee #3 (Remarks for Author):

The MS by Wilcz-Villega et al. proposes an interesting correlation between obesity, macrophage secreted factors, IKBe expression, the serine synthesis pathway and mammary tumorigenesis. The experiments are original and sound, and mostly support the conclusions. However, several points need to be addressed as follows:

1. The title is misleading. As only the expression of IKKBe correlates with inflammation in mammary tumours, TBK1 should not be mentioned in the title.
2. Introduction/Discussion section: please specify either here or in the Discussion whether the macrophages detected in the mammary gland of obese individuals are more of the M1 or M2 phenotype. Moreover, detail where increased angiogenesis has been detected in correlation with obesity-associated inflammation. Finally, for the sake of data interpretation, the authors should discuss the known role of IKBe in the hypothalamus in insulin resistance upon HFD.
3. Please mention in the results section how are MF derived and activated. Moreover, the polarized phenotype should be validated, for example by staining for CD68, CD80 and CD163, and by measuring the production of GM-CSF and M-CSF. In the same vein, a panel of relevant cytokines rather than just TNF α and CCL22 as in Fig. EV1 C,D have to be measured in the different MF populations obtained, especially since different results are obtained with the different CM.
4. Fig. 1A, EV2A: it would be interesting to assess the effects of the MF-derived CM also on primary mammary epithelial cells, as well *in vivo* growth of MCF10 cells pre-treated with MF-derived CM.
5. Fig. EV1 C,D: please add statistical analysis.
6. The detected difference between CM from M1A and M1D are interesting and would warrant a deeper investigation, such as for example comparing the two secretomes.
7. Fig 1D,E,F: if M1-CM, but not M2-CM, can increase proliferation in spheroids, then one would expect a difference in the degree of filling induced by M1 or M2, respectively. Please compare the two groups and calculate statistical significance. If no difference is detected, an alternative explanation is that only migration and not proliferation is involved in spheroids filling. This idea should be tested by inhibiting proliferation.
8. Fig. 1 G-E: The number of protrusions per sphere should be quantified.
9. Fig. EV2 G,H: M1A-CM reduces both migration and proliferation (see before). Please comment.
10. The last sentence of page 7 should be somehow split as at this stage no correlation is established between the effects of the MF-CM on mammary epithelial cells and the effects of HFD on invasive properties of mouse organoids. Even better, leave any comment for the Discussion.
11. Table I: the inflammation score is not sufficient; F4/80 should be measured by IHC.
12. Unpublished data cannot be cited as "previously shown"; please modify such as, for example: "we have recently observed that (Xu et al, submitted for publication, or unpublished data).

13. The first set of experiments on PyMT mice (Fig. EV5), is conceptually flawed, since this is a very aggressive model and such a big age difference within the analysed mice (9-13 weeks old) makes impossible any kind of comparison. Please eliminate from manuscript.

14. The design of the second set (Fig. 7) is correct. However, the differences are very subtle, and this is not surprising given the aggressiveness of the model. The choice of a less aggressive model such as the NeuT mice could potentially make these results much more relevant.

15. Last sentence in Discussion: the involvement of the SBP has also shown in this MS, and only observed in unpublished/submitted work (see above). Please correct.

1st Revision - authors' response

16 October 2019

1) At least the key experiments performed with a second cell line, or even better with primary cells. We used MCF10A cells since it is the only non-tumorigenic breast epithelial cell line that forms acini-like structure (spheroids) with a clear lumen when cultured in 3D for two weeks, recapitulating numerous features of the breast glandular architecture. This model allowed us to evaluate the consequences of macrophage-conditioned medium on filling of the spheroid lumen, recapitulating ductal carcinoma in situ, and the formation of invasive protrusions. We further confirmed our findings in a more complex model, i.e. mouse mammary organoids, and validated that macrophage conditioned medium promotes invasiveness. Thus, as you suggested, we have used primary cells in our study to confirm that macrophage-conditioned media promotes an invasive phenotype of the breast epithelium. In addition, we propose to perform the following experiments to further validate our findings:

1. Repeat the soft agar assay with further two cell lines: (i) N1089 – immortalised myoepithelial cell (Allen et al, 2014) and (ii) HMEC-M as suggested by Referee #2. For this experiment we plan to use macrophage conditioned medium derived from at least 3 new donors and to use both inhibitors, amlexanox and NCT502.

2. In collaboration with Richard Grose (at our institute), we will test luminal filling induced by macrophage-conditioned medium using their newly developed system (Carter et al, 2017). Briefly: myoepithelial and luminal cells isolated from two patients will be grown in collagen gels forming bilayer structures, resembling the breast architecture and macrophage conditioned medium (M1 activated and M2 activated only) derived from two donors will be applied in presence of amlexanox and NCT502.

1. We have repeated growth in soft-agar using the HMEC-M and N1089 the cell lines and macrophage conditioned medium derived from three donors (D18, D21, D22 and D17, D18, D21 respectively) (Rebuttal Fig.1). These two cell lines behaved very differently in soft agar in comparison to CF10A.

Indeed, control cells were already capable of growing in an anchorage independent manner forming on average 180 colonies (bigger than 50µm) per cm² in the case of HMEM cells (Rebuttal Fig1A) and 250 in the case of N1089 cells (Rebuttal Fig1D) in comparison to 2/3 colonies per cm² formed by MCF10A (Fig1A). Thus, it is not surprising that we did not see a clear effect in inducing colony formation by macrophage-conditioned medium, despite the media contained high concentration of TNFα (M1A) and CCL22 (M2A) (Fig. 1EVC-D). For this reason we decided not to add this data to the original manuscript, since we have now additional data using primary human cells confirming the transforming effect of macrophage-conditioned medium (see below). Of note, both Amlexanox and NCT502 reduced the number of colonies in both cell lines in all conditions. These data confirm the role of macrophage-conditioned medium in promoting transformation of normal epithelial cells but also suggest that an inflammatory microenvironment will not affect cells once they are already capable of anchorage-independent growth.

2. More importantly, the malignant phenotype induced by M1A or M2A conditioned media was recapitulated in a model system where primary human myoepithelial and luminal cells, isolated from reduction mammoplasty, are grown in collagen and reform into bilayer structures (Fig 2G-H). Indeed, using this recently established system (Carter et al, 2017) we could confirm that macrophage

conditioned media induce filling of the spheroid lumen recapitulating ductal carcinoma in situ, as we previously observed using MCF10A cells (Fig 1B-C). Importantly, amlexanox and NCT502 had a strong inhibitory effect also in this model system (Fig 4I-J and 6I-J).

2) At least the key experiments performed with genetic inhibition, to strengthen pharmacological inhibition. To address this point, we will generate MCF10A cells knockout for IKKe, PHGDH and PSAT1 by CRISPR/CAS9. These new cell lines will be used to test the effects of macrophage conditioned medium (M1 activated and M2 activated only) using the following readouts: (i) invasion in Matrigel/collagen (ii) filling of spheroid lumen and (iii) growth in soft agar. All the experiments will be performed using macrophage-conditioned medium derived from at least 3 new donors and both inhibitors – amlexanox and NCT502.

Please note this set of experiments might be limiting in time. It is difficult to predict how long it will take to generate the CRISPR cell lines and the soft agar experiment requires 5 weeks of culture.

As agreed, we successfully generated MCF10A CRISPR-Cas9 cell lines for IKKe and PHGDH (Fig EV5) that were used to strengthen the data coming from the use of pharmacological inhibitors. As shown in Fig 4K-L and 6K-L, deletion of IKBKE (coding for IKKe) or PHGDH genes perfectly recapitulated the phenotypes observed with amlexanox and NCT502 in terms of reducing growth in soft agar and inhibiting the formation of invasive protrusion induced by macrophage-conditioned medium.

3) A careful characterization of macrophage polarization and secreted cytokines/chemokines. Although the aim of our manuscript is not to establish a new protocol to generate macrophages to the currently available numerous methods, to which we refer in our manuscript, we do appreciate that some further characterisation of these cells is important. We will:

1. Perform flow cytometry to determine the expression of the following markers: HLA-DR, CD163, CD86, CD206 as in Georgouli et al (Georgouli et al, 2019).

2. Perform the following cytokine array: <https://www.abcam.com/cytokine-array-human-cytokineantibody-array-membrane-42-targets-ab133997.html>

1. Macrophages derived from donors 22-25 were characterised as described in (Georgouli et al, 2019) using four markers: CD206 that was described to be induced upon IL-4 treatment, CD86 by IFN γ and LPS, CD163 by IL10 and HLA-DR that was induced similarly by all treatments. We observed that the majority of the cells in the four populations were double positive for HLA-DR/CD86 (Fig EV1K-L), with M1D/A expressing HLA-DR at higher level than M2D/A, while CD86 was higher in M1A and M2A than in M1D and M2D (Fig EV1O-P). The percentage of cells double positive for CD206/CD163 was higher in the M2D/A populations (Fig EV1M-N) and CD206 was expressed at higher level by M1D and M2A, while CD163 by M2D/A (Fig EV1R-S). This characterization of the four cell populations confirmed that some markers attributed to M1 macrophages are indeed expressed by IFN γ and LPS treated cells (i.e. CD86) or CD206 by IL-4 treated cells. However, we also observed some differences between our data and what was reported in (Georgouli et al, 2019), such as HLA-DR being expressed at higher level in the M1D/A populations and not in all groups.

2. We also characterised secretion of the four populations from three independent donors (D19, D22 and D25) using an array detecting 42 cytokines/chemokines. All the results have been quantified and are shown in Table1. Fig EV1E shows a representative blot of the array done with medium conditioned by M1A cells derived from D25. Fig EV1F-J show some representative cytokines secreted by the three donors and confirm for example secretion of MIG and RANTES, generally attributed to IFN γ and LPS treated cells and TGF- β and IL-10 generally associated with IL-4 treated cells (Murray et al, 2014).

4) A careful comparison of statistically significant differences among the groups where indicated.

Yes, we will address this point.

Where relevant, this has been calculated and indicated in the figures.

5) *Improving in vivo experiments must at least be attempted: the referees recognize that a second transgenic model may take too long to establish, but a xenograft approach could be used (better in a syngeneic mouse model).*

We do appreciate the referees' concerns about the in vivo model and the different suggestions. However, it is important to highlight that our aim was to look at initial tumour formation, thus the in vitro experiments performed aimed to look at malignant transformation. For this reason, we think that a xenograft model will not be appropriate, since in these models already transformed cells are injected. Alternative considerations are (i) using NMuMG cells, however their syngeneic recipient NAMRU mouse is no longer available; or (ii) injecting our transformed MCF10A cells into mice. However, for this experiment we would need immunodeficient mice, which would not allow studying chronic inflammation (as in the case of HFD).

Thus, following discussion with our in vivo expert collaborator Prof Hodivala-Dilke we had to conclude that only another transgenic model would be appropriate, but as recognised by you and the referees, this would take too long.

As an alternative, to further strengthen our data we could propose to analyse all the tumours collected during our in vivo study and look for (i) pathological characterisation and (ii) macrophage infiltration.

As agreed, we analysed the tumours that were collected during our in vivo work. The following parameters were analysed by Alaistar Ironside, a breast cancer pathologist: (i) tumour stage, (ii) nuclear polymorphism, (iii) mitotic rate, (iv) differentiation grade. We also analysed F4/80+ macrophage infiltration at the (v) border and (vi) core of tumours. For this analysis we chose one tumour per mouse (smaller than 500 mm³) that was detected at least two weeks before the mouse was culled and it was at the time closest to 4 weeks, which was the average for all tumours across all groups. We also evaluated (vi) the total volume of tumours detected in each mouse. All these data are collected in Rebuttal Fig 2 and showed that the HFD did not affect any of these parameters and consequently amlexanox did not have an effect. These data are currently not in the manuscript, however we would be happy to add them if required.

6) *The first in vivo experiment should be removed (point 13 from referee #3).*

Yes, we will address this point and remove the first in vivo experiment.

This has been done. We removed the figure and the text has been amended accordingly.

References

Allen MD, Thomas GJ, Clark S, Dawoud MM, Vallath S, Payne SJ, Gomm JJ, Dreger SA, Dickinson S, Edwards DR, Pennington CJ, Sestak I, Cuzick J, Marshall JF, Hart IR & Jones JL (2014) Altered microenvironment promotes progression of preinvasive breast cancer: Myoepithelial expression of avb6 integrin in DCIS identifies high-risk patients and predicts recurrence. *Clin. Cancer Res.* 20: 344–357

Carter EP, Gopsill JA, Gomm JJ, Jones JL & Grose RP (2017) A 3D in vitro model of the human breast duct: A method to unravel myoepithelial-luminal interactions in the progression of breast cancer. *Breast Cancer Res.* 19: 1–10

Georgouli M, Herraiz C, Crosas-Molist E, Fanshawe B, Maiques O, Perdrix A, Pandya P, Rodriguez-Hernandez I, Ilieva KM, Cantelli G, Karagiannis P, Mele S, Lam H, Josephs DH, Matias-Guiu X, Marti RM, Nestle FO, Orgaz JL, Malanchi I, Fruhwirth GO, et al (2019) Regional Activation of Myosin II in Cancer Cells Drives Tumor Progression via a Secretory Cross-Talk with the Immune Microenvironment. *Cell* 176: 757-774.e23 Available at: <https://doi.org/10.1016/j.cell.2018.12.038>

Murray PJ, Allen JE, Biswas SK, Fisher EA, Gilroy DW, Goerdts S, Gordon S, Hamilton JA, Ivashkiv LB, Lawrence T, Locati M, Mantovani A, Martinez FO, Mege JL, Mosser DM, Natoli G,

Saeij JP, Schultze JL, Shirey KA, Sica A, et al (2014) Macrophage Activation and Polarization: Nomenclature and Experimental Guidelines. *Immunity* 41: 14–20 Available at: <http://dx.doi.org/10.1016/j.immuni.2014.06.008>

2nd Editorial Decision

11 November 2019

Thank you for the submission of your revised manuscript to EMBO Molecular Medicine. We have received the referees' reports, and as you will see they are now supportive of publication of your study. I am therefore pleased to inform you that we will be able to accept your manuscript pending final editorial amendments.

I look forward to reading a new revised version of your manuscript as soon as possible.

***** Reviewer's comments *****

Referee #1 (Comments on Novelty/Model System for Author):

The revised manuscript contains significant new data generated in response to prior review. These data greatly enhance the conclusions.

Referee #1 (Remarks for Author):

The flow cytometry bivariate plots presented in Fig EV1 are not acceptable. The quad plots should have straight lines that define the quadrants, which are rigorously set by negative controls. Please correct.

The additional new data represent a lot of effort in response to prior concern and have significantly improved the strength of the conclusions.

Referee #2 (Comments on Novelty/Model System for Author):

Compared to the original submission, the authors have made two major improvements:

Tested primary human myoepithelial and luminal cells, which significantly strengthen the observations made with cell lines.

Used genetic approaches to confirm pharmacological data, with consistent results.

Characterized the surface phenotype and the cytokine / chemokine secretion of the four populations on monocytes

Referee #2 (Remarks for Author):

This reviewer has no further questions or comments.

Referee #3 (Remarks for Author):

The MS has been reviewed according to suggestions and is now much improved.

2nd Revision - authors' response

3 December 2019

Authors made the requested changes.

Corresponding Author Name: Katuscia Bianchi

Manuscript Number: EMM-2019-10491-V2